# Impact of subsurface crevassing on the depth-age relationship of high-alpine ice cores extracted at Col du Dôme between 1994 and 2012

Susanne Preunkert[1,2], Pascal Bohleber[2,3,4] Michel Legrand[1,5], Adrien Gilbert[1], Tobias Erhardt[6,7], Roland Purtschert[6], Lars Zipf[2,8], Astrid Waldner[2], Joseph R. McConnell[9], Hubertus Fischer[6]

[1]Université Grenoble Alpes, CNRS, Institut des Géosciences de l'Environnement (IGE), Grenoble, France
[2]Institute of Environmental Physics, Heidelberg University, Heidelberg, Germany
[3]Institute for Interdisciplinary Mountain Research, Austrian Academy of Sciences, Innsbruck, Austria
[4]Ca' Foscari University of Venice, Department of Environmental Sciences, Informatics and Statistics, Scientific Campus, via Torino 155, 30172 Mestre (VE), Italy
[5]Laboratoire Interuniversitaire des Systèmes Atmosphériques, Université de Paris and Univ Paris Est Creteil, CNRS, LISA, F-75013, France
[6]Climate and Environmental Physics, Physics Institute, and Oeschger Centre for Climate Change Research, University of Bern, Switzerland
[7]Alfred Wegener Institute, Helmholtz Centre for Polar and Marine Research, Bremerhaven, Germany
[8]Laboratoire de Glaciologie, Université Libre de Bruxelles, Brussels, Belgium
[9]Division of Hydrologic Sciences, Desert Research Institute, Reno, Nevada, USA

*Correspondence to*: Susanne Preunkert (susanne.preunkert@iup.uni-heidelberg.de)

**Abstract.**

Three seasonally resolved ice-core records covering the 20[th] century were extracted in 1994, 2004 and 2012 at a nearly identical location from the Col du Dôme (4250 m above sea level, m asl, Mont Blanc, French Alps) drill site. Here we complete and combine chemical records of major ions and radiometric measurements of $^3$H and $^{210}$Pb obtained from these three cores together with a 3D ice flow model of the Col du Dôme glacier to investigate in detail the origin of discontinuities observed in the depth-age relation of the ice cores drilled in 2004 and 2012. Taking advantage of the granitic bedrock at Col du Dôme, which makes the ice core $^{210}$Pb ice-core records sensitive to the presence of upstream crevasses, and the fact that the depth-age disturbances are observed at depths for which absolute time markers are available, we draw an overall picture of a dynamic crevasse formation. This can explain the non-disturbed depth-age relation of the ice core drilled in 1994 as well as the perturbations observed in those drilled in 2004 and 2012. Since crevasses are common at high alpine glacier sites, our study points to the important need of rigorous investigations of the depth-age scale and glaciological conditions upstream of drill sites before interpreting high alpine ice core records in terms of atmospheric changes.

## 1. Introduction

Close proximity to European source regions makes ice cores from high-elevation Alpine glaciers an important target to reconstruct past anthropogenic perturbations of atmospheric chemistry. In the French Alps, the Col du Dôme (CDD) glacier close to the Mont Blanc summit has been studied extensively over the last 25 years for its glaciological properties and suitability for glacio-chemical studies (e.g. Vincent et al., 1997, Preunkert et al., 2000). Although it has experienced significant warming in response to climate change since the 1980s (Vincent et al., 2007; Gilbert and Vincent, 2013; Vincent et al., 2020), the glacier has been shown to be entirely cold (i.e. the ice temperature is below freezing point at all depths), with the exception of sporadic surface melting and refreezing in the uppermost centimeters during summer. Ice cores extracted at CDD have been used to reconstruct various aspects of atmospheric changes during the 20$^{th}$ century over western Europe. These include major inorganic species ($NH_4^+$, $NO_3^-$, and $SO_4^{2-}$, Fagerli et al., 2007, Preunkert et al., 2003, and Preunkert et al., 2001a), halogens (HCl and HF, Legrand et al., 2002, Preunkert et al., 2001b; total I and Br, Legrand et al., 2018 and Legrand et al., 2021), black carbon (Moseid et al., 2022), dissolved organic carbon (DOC, Legrand et al., 2013), organic molecules (Legrand et al., 2003 and 2007, Guillermet et al., 2013), and trace elements such as Pb and Cd (Legrand et al., 2020), V and Mo (Arienzo et al., 2021), and Tl (Legrand et al., 2022). Underpinning these efforts are three ice cores all drilled to bedrock within at most 10 m of each other (mean geographic location of 45.842195° N, 6.84675° E) in 1994 (C10, Vincent et al., 1997, Preunkert et al., 2000), 2004 (CDK, Legrand et al., 2013) and 2012 (CDM, Legrand et al., 2018, this study).

For the C10 ice core, drilled in 1994 a depth-age relationship (Fig. 1) was derived that was consistent for annual layer counting and several time markers. The age scale is strictly monotonic, meaning it does not show evidence of flow anomalies, hiatuses, or folding within the dating uncertainty (Preunkert et al., 2000). In contrast, it was shown that the 2004 CDK core is likely missing ~16 years between ~1970 and ~1954 (Legrand et al., 2013) as confirmed by the absence of the well-known $^3$H maximum in 1963 caused by atmospheric nuclear tests. Although its precise cause remained unclear, it was suggested that the missing 1954-1970 period was related to an (upstream) crevasse that had disturbed the continuity of the CDK record through inflow of a partly snow filled crevasse to the ice core site. The presence of one or more crevasses in the upstream vicinity of the drill site also was suspected to cause strongly elevated concentrations of $^{210}$Pb observed in the C10 core (Vincent et al., 1997) because the bedrock at the CDD consists of granite that emits $^{222}$Rn (half-life of 3.8 days). $^{222}$Rn is able to diffuse in snow and firn, but much less so in ice (see also Pourchet et al., 2000), and subsequently decays to produce $^{210}$Pb (half-life of 22.3 years). The existence of a $^{210}$Pb anomaly despite the continuous and monotonic depth age relationship of core C10 suggests that the crevasse was close but did not intersect the C10 flow line or that the crevasse closed again without substantial disturbance of the ice stratigraphy in the decades prior to 1994 (see discussion below for more details).

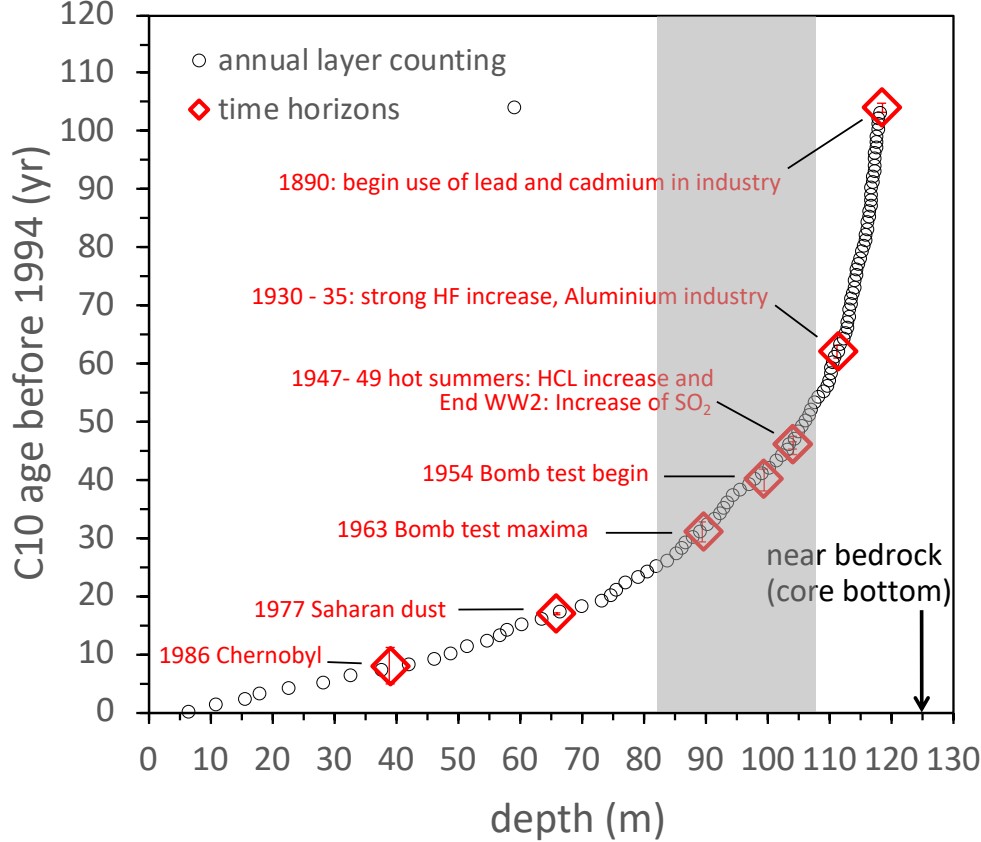

**Figure 1: Depth-age relation established for C10 between 2000 and 2016. Data are from Preunkert et al., 2000, Vincent et al. 1997, Preunkert et al., 2001a, Preunkert et al., 2001b, Legrand et al., 2002, Legrand et al., 2018 and Preunkert et al., 2019a and the age scale is based on annual layer counting and absolute age markers. The shaded area refers to the depth zone where enhanced $^{210}$Pb values (see Sect. 3.2 and Fig. 6) were observed**.

For the 2012 CDM core, only the upper core sections (down to 81 m depth, i.e. 1979) have been investigated previously for various trace elements including major ions, black carbon, halogens, Pb, Cd, V and Mo, Tl, Bi and P (Legrand et al., 2018, 2020, 2021, 2022, Arienzo et al., 2021, Moseid et al., 2022, Eichler et al., 2023, Legrand et al., 2023a, 2023b). Here we report additional $^{210}$Pb measurements, and use $NO_3^-$, $NH_4^+$ and $^3$H analysis to extend the depth-age relationship of the CDM core back to 1950. This homogeneous set of chemical and radiochemical data from C10, CDK and CDM ice cores permits investigation of the consistency of the depth-age relation back to 1950 between these ice cores drilled in 1994, 2004, and 2012, respectively. In addition, a first attempt to provide a qualitative glaciological explanation for the observed discontinuity in the depth-age relation of CDK and CDM and the link with the presence of unexpectedly high $^{210}$Pb levels will be made. This is important for understanding the extent to which existing and future ice cores drilled at this location on the CDD saddle are suitable to reconstruct past atmospheric chemistry changes.

## 2. Site and Analysis

### 2.1 Site characteristics

The CDD site is located on a small cold glacier saddle downslope of the Dôme du Gouter (4300 m asl) (Fig. 2). On this slope, the C10, CDK, and CDM cores were drilled down to ~125 m (Table 1), i.e., close to bedrock. Detailed glaciological descriptions of this site can be found in Vincent et al. (1997, 2020), whereas Preunkert et al. (2000) characterized this site in terms of its usefulness to reconstruct past atmospheric changes since the beginning of the 20[th] century based specifically on data from the C10 ice core. Ice flow, firn compaction and thermal regime have been modeled in three dimensions by Gilbert et al. (2014), allowing particle back-tracking and flow-based estimation of the depth-age relationship for the drilling site. However, Gilbert et al. (2014) did not allow for the occurrence of crevasses along the slope from Col du Gouter to the Col du Dome saddle. In fact, field observations and photographic evidence shows the existence of a large crevasse (clearly visible by a depression at the surface although the crevasse is not necessarily open to the atmosphere) east of the CDD dome which, dependent on its north-south extension, could also intersect the upstream flow line from the drill site. Unfortunately, we do not have direct measurements of the depth and lateral extent of the crevasse. The crevasse appears approximately at an oversteepening of the bedrock topography (Fig. 2c) along the flow line, suggesting that it is extensive stress at the bottom that leads to crack formation at a specific point and allows for opening of a deep crevasse down to bedrock (in line with $^{210}$Pb evidence as outlined below). We stress that the crevasse is not necessarily open to the surface (see Figure 2 a and b), but that collapse of the snow bridge at the top in the past cannot be ruled out. We also note that this crevasse is not moving downhill with the surface velocity of several meters per year in the observations but is found approximately at the same location of the glacier surface every year. Despite this stationarity of the crevasse, the surface velocity field is not disturbed (Gilbert et al. 2014) implying that the ice flow is not totally interrupted across the crevasse. Together with the stationarity of the crevasse, this suggests that the subsurface void created by the crevasse is filled again by glacier flow after its opening (as also suggested by significant glacier thickness reductions of a few meters from 1993 to 2017 (Vincent et al. 2020) in the vicinity of the crevasse). Accordingly, we interpret the glaciological evidence as (recurrent) opening but also potential re-closure of the crevasse (Colgan et al., 2016) below the surface at the same bedrock topography-induced position. Comparing photos taken in 2012 (Fig. 2a) and in 1999 (Fig.2b), shows widening and northward extension of the crevasse from 1999 to 2012. Whereas in 2012 the crevasse is clearly visible as a snow-covered depression on the surface slope, the crevasse appeared to be limited to the southwestern flank of the drill site catchment area in 1999. Following Fig. 3, the crevasse is situated more than 100 m upstream of the drill site of C10, CDK, and CDM. Figure 3a shows the CDD glacier thickness changes between 1993 and 2017 overlayed with the modelled flow line indicating the calculated arrival depths at the drill site of C10, CDK, and CDM (Gilbert et al., 2014). Figure 3b and c represent vertical cross sections along the modelled flow line in Fig. 3a overlayed by simplified sketches of the upstream crevasse visible in Fig. 2. We sketch the crevasse in two hypothesized temporal states, as concluded in Sect. 4 on the basis of C10, CDK and CDM ice core data presented in Sect. 3. Table 1 summarizes the main characteristics of the three ice cores and basic findings related to radiometric analyses.

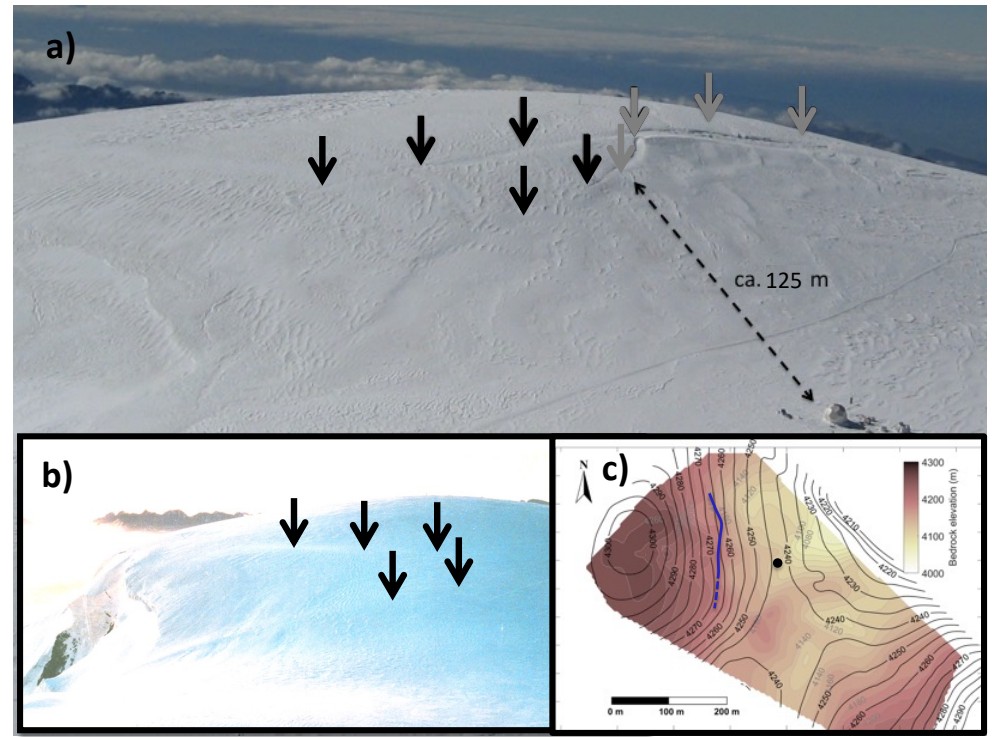

**Figure 2: View of the South-East flank of the Dome de Gouter and Col du Dome saddle including the drill site of 1994, 2004, and 2012 situated downslope of Dome du Gouter. Note that the three drill sites are located within about 10 m of each other and thus are indicated by a single dot in (c). (a) Picture taken in summer 2012: A large crevasse extends across the upstream catchment area of the drilling site. At the time of the picture the distinctly visible crevasse was mainly snow-covered. A potential second crevasse also is visible on the southwestern slope of the glacier. (b) Picture taken in summer 1999: Evidence of one to two crevasses limited to the southwestern side of the Dome du Gouter. Black arrows in a) and b) indicate parts of the crevasse which are suggested by the surface features in a) and b). Grey arrows in a) mark the part of the crevasse which was only visible in 2012 (c) Topographic map of the Col du Dome and Dome de Gouter together with the underlying bedrock topography (adapted from Vincent et al., 2020). Contour lines are spaced at 5 m intervals. The main crevasse highlighted in (a) and (b) is reported in (c) based on an aerial photo from Institut national de l'information géographique et forestière (IGNF) taken at 30th June 2004 (blue solid line in (c) indicates the part of the crevasse which was clearly visible, and blue dashed lines demark the part which was less clearly visible).**

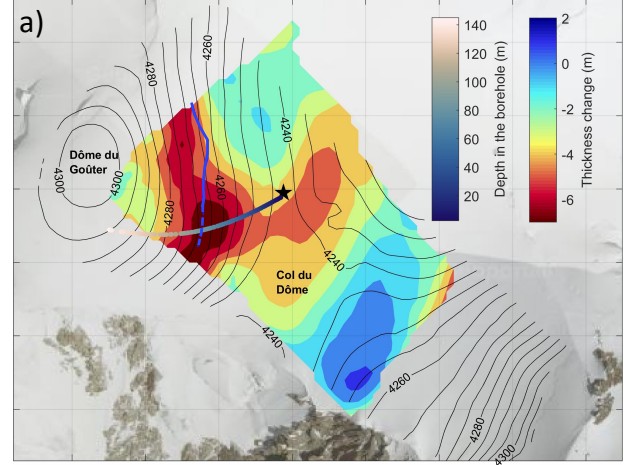

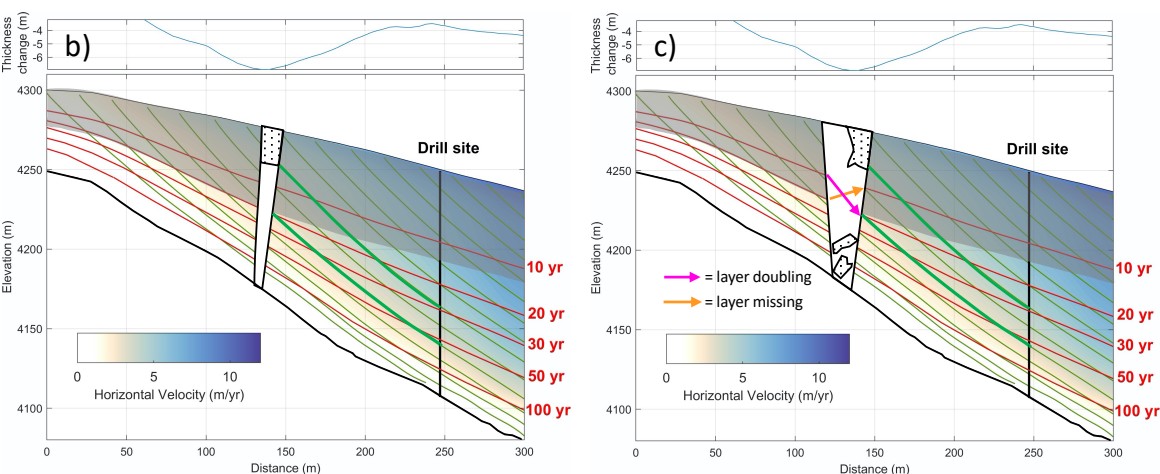

**Figure 3: (a) CDD thickness changes between 1993 and 2017. The contour lines of surface topography correspond to the 1993 surface (adapted from Vincent et al., 2020) overlain by a modelled flow line (color scale on top) which reports the calculated arrival depth**
**at the drill site of C10, CDK, and CDM (black star) (Gilbert et al., 2014) Note the direction of the flow line which is not perpendicular to the surface topography, but is mainly controlled by the bedrock topography. The crevasse location (blue line) is based on the 30th June 2004 aerial photo from IGNF (see Fig. 2) (b and c) Schematic representation of the origin of the [210]Pb anomalies found at the drill site following the ice flow model of Gilbert et al., 2014, extracted along the flow path reaching the drill site. The crevasse is indicated approximately at its observed upstream position but its width is not drawn to scale. Isochrones are marked in red, flowlines**
**in green (see also Sect. 4). The grey shaded zone indicates firn, the dotted zone indicates the snow bridge over the crevasse. Note that the two thick green flow lines enclose approximately the depth section in the ice cores affected by the [210]Pb anomalies. The sketch shows that the upstream origin at the crevasse location of the lower flow line coincides with the firn close-off depth. The upper flow line originates higher up in the firn at the crevasse location, here hypothesized to coincide with the lower end of the snow bridge. Based on our conclusions derived from the C10, CDK and CDM ice core data (see Sect. 3 and 4), two states of the crevasse are**
**suggested: (b) in the years ~1965-1970 (i.e. ~25-30 years before the C10 drilling) the crevasse is open to the bedrock but sealed from the atmosphere by a snow bridge. In this state [222]Rn and [210]Pb accumulate and reach concentrations well above atmospheric conditions in the crevasse and the surrounding firn. (c) after ~1975 and at least until ~1990 (i.e. ~25-30 years before the CDK and CDM drilling), the crevasse could have been partly open to the atmosphere. In this state [222]Rn and [210]Pb concentrations in the crevasse and the surrounding firn are strongly reduced compared to (b). Due to the infill of the crevasse by glacier flow and snow**
**from above either missing or doubling of ice layers in the ice flowing downhill is possible as indicated by the orange and pink arrows (see also main text).**

**Table 1: Basic glaciological and radiometric parameters of the three CDD ice cores investigated in this study.**

| Core name | C10 | CDK | CDM |
|---|---|---|---|
| Drilling year | 1994 | 2004 | 2012 |
| Ice core length [m] | 126 | 124 | 122.5 |
| Surface (uppermost 15years) accumulation [m (mwe)] | 6 (2.6) | 3.8 (2.5) | 3.5 (2.3) |
| Accumulation over uppermost 30 years [m (mwe)] | 2.9 (2.1) | 2.7 (2.0) | 2.6 (1.85) |
| Firn-ice transition [m (years)] | 56 (13) | 54 (14) | 52 (14) |
| Depth of the $^3$H maximum [m] | 87.67 | - | 93.3 (87.3[a]) |
| Top and bottom depth of the $^{210}$Pb anomaly [m] | 83–108 | 85–~108 | 81-~102 |

[a] Depth of shallower $^3$H maximum detected in CDM, considered as not corresponding to the year 1963 (see Sect. 3.1.3)

## 2.2 Ice core analysis

Table 2 reports the analytical methods of ice core analysis used within this study. $^3$H analyses in CDK (Legrand et al., 2013) and CDM ice were performed at the Institute for Environmental Physics, Heidelberg University (IUP), by low-level gas counting with a detection limit typically around 1.5 TU (tritium units) and a mean measurement error (depending on measurement time and sample size) for the sample set presented here of $1.5 \pm 1.2$ TU. $^3$H analyses using liquid scintillation counting with a mean error of $2.5 \pm 2.7$ TU (for the here presented samples) in CDM ice also were performed at the Division for Climate and Environmental Physics (CEP) of the Physics Institute, University of Bern, using ice core samples at higher depth resolution than used at IUP.

$^{210}$Pb samples of CDK and CDM ice were analyzed at IUP by α-spectrometry for its decay product $^{210}$Po. Typical blank values of $(5.7 \pm 2.5)\ 10^{-5}$ Bq for $^{210}$Po and $(3.8 \pm 1.6)\ 10^{-5}$ Bq for $^{209}$Po were subtracted from the sample counts (see Stanzick, 2001, and Elsässer et al., 2011 for further analytical details). With typical sample masses of 300 to 1,000 g and measurement times of 2 to 6 days, mean $^{210}$Pb measurement errors of $4 \pm 4$ mBq kg$^{-1}$ were achieved on ice core drill chip samples spanning ice core depths between 0.6 and 1 m. Previously reported $^{210}$Pb measurements in C10 ice (Vincent et al. 1997) analyzed at the Laboratoire de Glaciologie et Géophysique de l'Environnement (now Institut des Géosciences de l'Environnement (IGE)) were complemented by two additional samples. The analytical technique used was high-resolution gamma-ray spectrometry, designed to detect very low levels of radioactivity using a 20% high-purity Ge (N-type) detector, with an anti-Compton scintillation detector (Pinglot and Pourchet,1995) for which snow and ice samples were filtered previously through ion-exchange papers (Delmas and Pourchet,1977). We note that the gamma-ray method is less sensitive than α-spectrometry due to the high conversion of the low energy γ-line at 46 keV (96% in the form of electron and only 4% in the form of γ emission) (Gaeggeler et al.2022), and may have systematic differences. Although made on a sediment sample with much higher

specific $^{210}$Pb activities than found in core cores, Pinglot and Pourchet (1995) made a direct comparison of $^{210}$Pb alpha and gamma-ray measurements. They found that the measurements made with α-spectrometry were only ~84 ± 11 % of the respective values obtained with gamma spectrometry and attributed the difference to insufficient acid leaching during the α-spectrometry sample preparation. However, both methods generally provide comparable activity values and the relative temporal variations in the activities should be robust. Vincent et al. (1997) did not assign uncertainties to their analyses. Here we estimate the uncertainty based on what has been reported in other studies using this detection method developed at IGE. Pinglot et al. (2003) reported a detection level of 10 mBq at a 97.5% confidence level for 3 days of counting on ice core samples with a typical $^{210}$Pb activity of 20 – 50 mBq kg$^{-1}$. These measurements included Chernobyl fallout in sub-Arctic glacier sites, and the levels were similar in range to the background activities of 50-100 mBq kg$^{-1}$ found in our cores. On the other hand, detection levels of 13 (and 25 mBq) were calculated at 97.5 % confidence when peak interferences where neglected (or considered), respectively, for a 10 g sediment sample containing 1000 times higher $^{210}$Pb activities as found in ice cores (~70 Bq kg$^{-1}$) and that was measured for 63 hours (Pinglot and Pourchet, 1995). Vimeux et al. (2008) reported a lower detection limit of 4 mBq kg$^{-1}$ for $^{210}$Pb measurements (activities between 20 and 100 mBq kg$^{-1}$) on relatively small (150-250 g) ice core samples from Patagonia.

The $^{210}$Pb activities in C10 ranged from 50 to700 mBq kg$^{-1}$, with the measurements done on the C10 drill chips merged over 3 to 5 m, allowing to obtain sample weights of ~3 to 5 kg. Since these sample masses, sample type (ice core sample) and geometry (filter) are comparable to those used in the Pinglot et al. (2003) study but are very different from the sediment sample in Pinglot and Pourchet (1995), we assume in the following a detection level of 10 mBq assigned by Pinglot et al. (2003) and a maximum uncertainty of 30 mBq for all C10 $^{210}$Pb measurements. Taking 1 kg sample mass as an absolute lower limit, this would amount to a total error of 30 mBq kg$^{-1}$. Note that, the dataset from Vincent et al. (1997) was complemented by two additional samples for which $^{210}$Pb analysis and quality control were not available in 1997. With measured activities of 760 and 460 mBq kg$^{-1}$ of $^{210}$Pb, these samples initially had been suspected to be contaminated and were not included by Vincent et al., 1997. Re-measurements of the respective ice core sections using samples extracted from the center of the core, however, confirmed the initial values so they must be considered as valid and were included in the data set of this study.

We stress that whereas $^{210}$Pb was measured continuously on discrete samples covering the whole C10 ice core, $^{210}$Pb measurements in CDK and CDM were focused on the $^{210}$Pb anomaly starting around 80 m depth. Therefore, only point-wise measurements with sample lengths between 0.6 to 1m length, i.e. covering less than one year, were made in the upper part of the latter two cores, with the exception of two CDM samples which were integrated of over core depths of 10 m each (covering 2 and 4 years).

Ionic species were analyzed continuously on discrete samples using ion chromatography along the upper 35 m of the CDM core at IGE (Eichler et al., 2023). Regular continuous flow analyses (CFA) on the CDM core, including nitrate (NO$_3^-$) and ammonium (NH$_4^+$), were made at Desert Research Institute (DRI) in Reno from 45 to 86 m depth (see Legrand et al. (2018) and references therein). Additional NO$_3^-$ and NH$_4^+$ data that are useful to derive an age scale by annual layer counting at CDD (Preunkert et al., 2000) were obtained on CDM ice with CFA measurements conducted at CEP along the whole ice core.

**Table 2: Analytical methods of ice core analysis used for the present study**

| Parameter | C10 | CDK | CDM |
|---|---|---|---|
| $^{210}Pb$ | gamma-spectrometry (IGE) (two samples, this study; all others Vincent et al., 1997) | alpha-spectrometry (IUP, Legrand et al., 2013) | alpha-spectrometry (IUP, this study) |
| $^{137}Cs$ | gamma-spectrometry (IGE, Vincent et al., 1997) | - | - |
| $^3H$ | - | gas counting (IUP, Legrand et al., 2013)) | gas counting (IUP, this study) / liquid scintillation (CEP, this study) |
| $NO_3^-$ and $NH_4^+$ | ion chromatography (Preunkert et al., 2003; Fagerli et al., 2007)- | ion chromatography (Legrand et al., 2013)- | continuous flow analyses (DRI, Legrand et al., 2018; CEP, this study) |

Details of the CFA analyses at CEP are provided in Kaufmann et al. (2008), Gfeller et al. (2014) and Erhardt et al (2022).

However, since the CDM ice core has only a 3-inch diameter, the ice available for the CFA analyses at CEP consisted only of a non-rectangular cross-section with maximum outer dimensions of 2.5 x 3.0 cm instead of the standard quadratic size of 3.5 x 3.5 cm for which the standard melt head at CEP is designed. Although a special, smaller melt head was constructed for the CDM analyses, it was not always possible to assure that the CFA melt water only came from an inner section of the ice material with no contact to the outer surfaces. This may have led to a higher risk of contamination of the inner sample melt water stream

and with the smaller melt water flow available implied also a reduced analyte spectrum. Despite the undersized core section available for the CFA analyses at CEP, 86% of $NO_3^-$ and/or $NH_4^+$ raw data could be evaluated. To test the reliability of the CEP dataset, the nitrate profiles obtained at DRI and CEP (covering 97% in this depth range) were compared over the depth interval 45 to 86 m. Both datasets are in very good overall agreement, except for individual outliers in the CEP data.

After having additionally discarded very high peaks (concentrations above 700 ppb) in $NO_3^-$ values (1.5% of CEP data in the

225 depth interval from 45 to 86 m), which were not present in the DRI dataset and could be attributed easily to contamination, mean $NO_3^-$ values over this depth interval were $263\pm281$ ppb (CEP) and $255\pm231$ ppb (DRI) (Fig. 4). The agreement is somewhat weaker for $NH_4^+$ likely because for this species only 80% of this depth range is covered by CEP measurements. After discarding additionally 8 % of the CEP $NH_4^+$ data between 45 and 86 m consisting of high $NH_4^+$ peaks (concentrations exceeding 190 ppb), which were not present in the DRI dataset, the mean $NH_4^+$ values of $101\pm110$ ppb (CEP) and $95\pm99$ ppb

(DRI) were in good agreement. As a consequence of the better reliability, we base our discussion mainly on the $NO_3^-$ data. Below 86 m no additional data were discarded from the CEP $NO_3^-$ and $NH_4^+$ datasets. However, because no further single $NO_3^-$ peak values above 700 ppb were found below 86 m, we are confident in $NO_3^-$ data below this depth. In the case of $NH_4^+$ we cannot exclude that a few peaks in the record below 86 m with a concentration higher than 200 ppb might be influenced by contamination.

## 3. Dating and $^{210}$Pb data

### 3.1 Ice Core Dating

The net annual accumulation in the upper layers of the drill site covering the upper 15 years is on average 2.5 m water equivalent (mwe) (Table 1) and typical at high alpine glacier sites (Vincent et al., 2020; Bohleber, 2019). The surface mass balance observed in the upstream area of the drilling site (i.e., upwind in the southeastern Dôme du Gouter flank, Fig. 2) decreases by one order of magnitude and reaches only ~0.2 mwe yr$^{-1}$ at the summit of the Dôme de Gouter (Vincent et al., 1997, 2020), where the glacier thickness is only ~40 to 45 m. We point out that due to the much lower accumulation rate in the Dome region, the firn-ice transition is also much shallower than in the lower outflow regions, where higher vertical velocities move this transition downward. In addition to the annual layer thinning caused by the glacier flow, the upstream net accumulation decrease, which is accompanied by a decrease of the winter to summer net snow accumulation rate, also impacts the annual layer thickness at the drill site. As a consequence, annual layer thicknesses of only 0.7 and 0.2 mwe are observed at 100 m and 118 m depth (Preunkert et al., 2000) and the winter to summer layer thickness ratio (which was calculated using the seasonal information embedded in the various aerosol tracers measured in the core, Preunkert et al., 2000), decreases from 1 at the surface to ~0.5 at 100 m depth.

Based on the well-marked seasonality in the chemical stratigraphy for all cores, annual layer counting was used as the main dating tool over the time period of interest of this study, i.e., back to the 1950s. NH$_4$$^+$ has a very strong seasonal variation (factor of ~14 higher in summer than in winter) caused by the parallel seasonal changes in source strengths and vertical transport of NH$_4$$^+$ (Preunkert et al., 2000). However also other ions (such as nitrate and sulfate) show clear seasonal variations (factor of ~4 higher in summer than in winter). The annual layer counting, which was based mainly on ammonia, was reinforced by absolute time markers such as Saharan dust events (for example the prominent event in 1977) (Preunkert et al., 2000 for C10; Legrand et al., 2013 for CDK, Legrand et al., 2018 and this study for CDM) and radiometric analyses aimed at detecting fallout from atmospheric thermonuclear bomb testing via $^3$H (Legrand et al., 2013 for CDK and this study for CDM) and $^{137}$Cs (Vincent et al., 1997) for C10. Fallout from atmospheric thermonuclear bomb testing typically leads to elevated $^{137}$Cs and $^3$H levels from 1954 to about 1975, with maxima in 1963 if the depth-age relationship is well preserved. The $^{210}$Pb depth profiles (Vincent et al., 1997 for C10) also were obtained in the three ice cores, but because of the presence of the strong anomalies discussed in Sect. 3.2, these data are not useful as dating tools.

### 3.1.1 The C10 core

The dating of the C10 ice core back to 1925 obtained by annual layer counting of the ammonium record and supported by absolute time markers was initially established by Preunkert et al. (2000). More recently, new measurements of toxic metals such as lead, cadmium and thallium underpinned identification of an additional absolute time marker, visible as a marked concentration increase in all three metals at the beginning of the industrial period, what allowed to extend the C10 chronology back to 1890 (Legrand et al., 2018). This additional information did not significantly change the original dating back to 1935

(i.e., only by one year back to depth 106.5 m and 5 years at a depth of 112 m), and these changes are within the estimated dating uncertainty of 5 to 10 years (Preunkert et al., 2000).

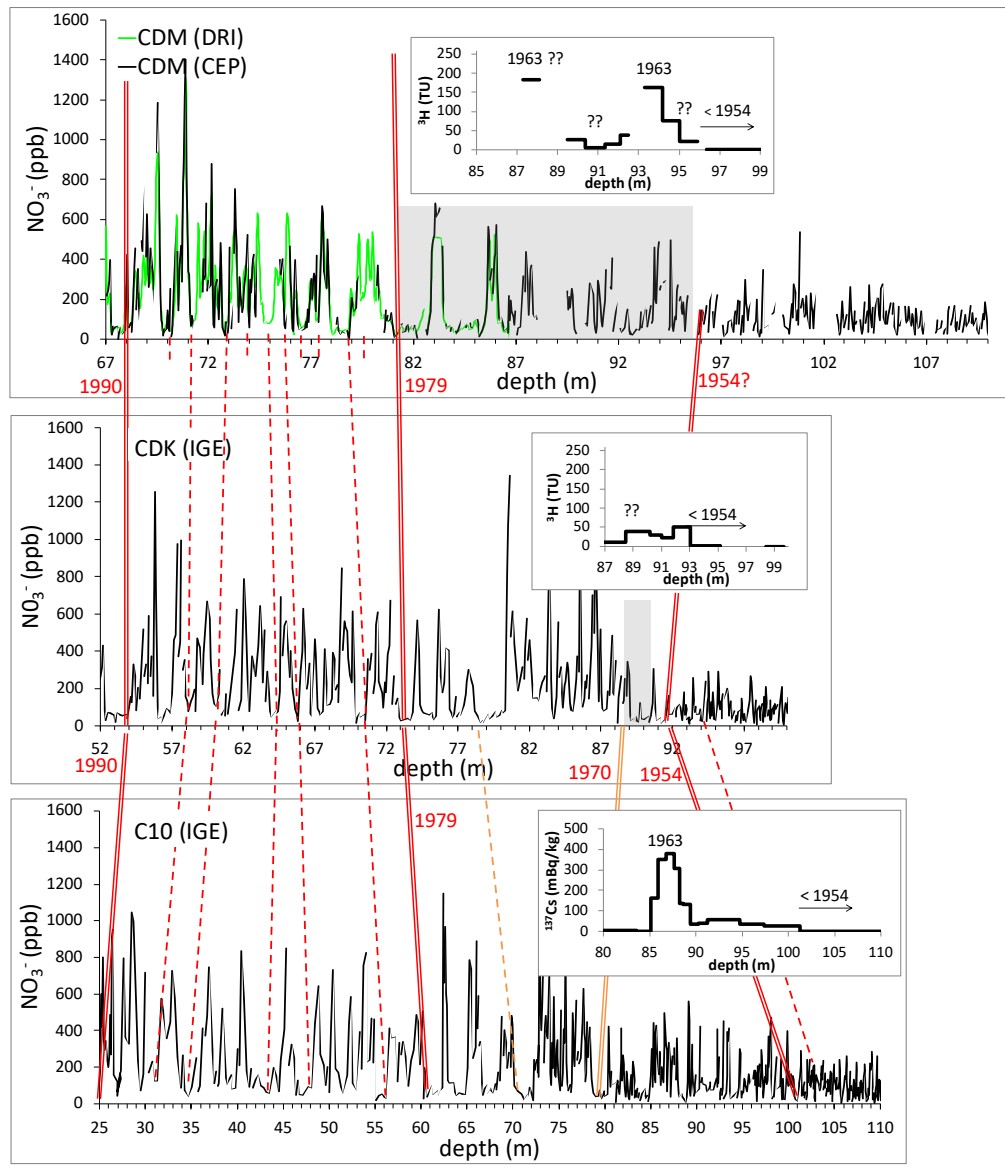

**Figure 4. Comparison of nitrate depth stratigraphies for CDM (this study), CDK (Legrand et al., 2013) and C10 (Preunkert et al., 2003). In addition, for each core the bomb test horizons in ³H are reported (this study for CDM, Legrand et al., 2013 for CDK and Vincent et al., 1997 for C10). Double and dashed red lines, indicate common years found in all three cores. Double and dashed orange lines assign common years found only in C10 and CDK. Finally, short red dashed lines mark annual layers in the upper undisturbed**
**part of CDM. Grey zones mark depth layers with apparent stratigraphic anomalies in CDM and CDK (see main text). Note that the chronological changes of the NO3 concentrations are offset in depth relative to each other due to the different years the cores were drilled.**

The updated C10 chronology is reported in Fig. 1. The agreement between results of the annual layer counting and several time markers in C10 shows that the depth-age relation is continuous and age increases monotonically with depth. The dating of the C10 core was found to be in excellent agreement with several pronounced atmospheric changes or events that occurred during the 20[th] century such as the $^{137}$Cs peak caused by nuclear weapons testing fallout (Vincent et al. 1997), the well-marked increase of fluoride after 1930 resulting from the rapid growth of the aluminum industry (Preunkert et al., 2001b), the large increase of sulfate after World War II (Preunkert et al., 2001a), and hydrochloric acid (HCl) peaks during the unusually hot summers from 1947 to 1949 caused by large forest fires (Legrand et al., 2002). Several of these events are recorded within the C10 depth interval where increased $^{210}$Pb values were observed (Vincent et al. 1997, Fig. 6 and Sect. 3.2). Thus, we can assume that the depth-age relation of the C10 core was not disturbed (i.e., by more than the dating uncertainty estimated to be ± 5yr, at a depth of 90 m (Preunkert et al., 2000)).

### 3.1.2 The CDK core

As done in the C10 core, the dating of the CDK ice core primarily was achieved by annual layer counting. However, the seasonality of ammonium as well as those of other major ions such as nitrate and sulfate cannot be discerned below 89.5 m, but recovers at depth larger than 92 m (Legrand et al., 2013). Since the CDK $^{3}$H profile does not show a clear bomb maximum of similar magnitude as in the other cores, Legrand et al. (2013) previously hypothesized that net snow accumulation corresponding to a few years around 1963 are missing due to the existence of an upstream crevasse, but suggested neither a reasonable glaciological mechanism for this effect or an explanation for whatever reason it appeared in CDK and not in C10. The comparison of ammonium, nitrate, and sulfate mean summer concentrations in the CDK core with those in C10 layers deposited above 89 m and below 92.0 m depth, however, suggests a reliable CDK record for the time intervals 2004 - 1970 and for a few decades older than ~1954 (see Legrand et al., 2013 and Fig. 4 and 5). Analogous to C10 (see Sect. 3.1.1), the CDK dating was updated in the lower part on the basis of additional measurements of trace metals such as lead and cadmium without changing significantly the original dating of Legrand et al. (2013) back to 1935 (Preunkert et al., 2019a).

A closer look at the NO$_3^-$ and NH$_4^+$ raw data in CDK shows that the depth interval from 80 to 89 m appears to correspond to 72 to 79 m depth in C10 and may be attributed to the years 1976-1971 as done by Legrand et al. (2013). However, these 5 years span 2 m more in CDK than in C10 corresponding to a relative thickening of layers by a factor of 1.28. Such an anomaly in the thinning curve cannot be explained by the systematic layer thinning at the drill site caused by undisturbed upstream inflow of ice (see also Fig. 3) along the same flow line for CDK and C10. However, refilling the void of the crevasse by inflow of ice from upstream may explain such a thickening.

### 3.1.3 The CDM core

Ionic species were analyzed as discrete samples using ion chromatography along the upper 35 m of the CDM core at IGE. From 45 m to 86 m depth, sections were measured using CFA at DRI (Legrand et al., 2018). These previous data were complemented by CFA measurements ($NO_3^-$ and $NH_4^+$) performed at CEP. Figure 4 shows sequences of the CDM depth-profile for nitrate in comparison with those from CDK and C10. Down to 81 m depth, the nitrate CDM stratigraphy matches very well those from C10 and CDK, where this depth is dated to 1979 based on annual layer counting. For the interval 1992-79 the annual layer thickness in CDM is approximately half that in CDK as expected, since the corresponding ice layers are deeper in the CDM core than in CDK (hence more thinned by glacier flow and deposited further upstream from the drill site). Below 81 m depth, the CDM depth profile differs from those of the two other cores. The three $NO_3^-$ peaks between 81 and 88 m depth in the CDM core (Fig. 4) could potentially be attributed to those seen between 73.2 and 78 m depth in CDK, but this would imply an annual layer thickness 1.75 times larger in the CDM than the CDK core. Again, this could only be explained by anomalous thinning (relative thickening of layers at great depth) caused by some crevasse-induced flow anomaly. In addition, the preceding assumption that the three nitrate peaks between 81 and 88 m depth in CDM date to 1978, 1977, and 1976, is in conflict with the $^3H$ level found in this core (see Fig. 4). Interestingly, whereas a winter to summer layer thickness ratio of ~0.55 is expected at ~80–100 m depth from the C10 core (see Sect. 3 and Preunkert et al., 2000), a very high winter to summer layer contribution (>2) is observed in the CDM core between 81 and 88 m depth. Such an unexpectedly high winter to summer contribution was also observed in CDK between 89.5 and 92 m depth, i.e. where the $NH_4^+$ seasonal cycle vanished. In summary, an anomaly in the ice stratigraphy occurs between 81 and at least 88 m depth in the CDM core.

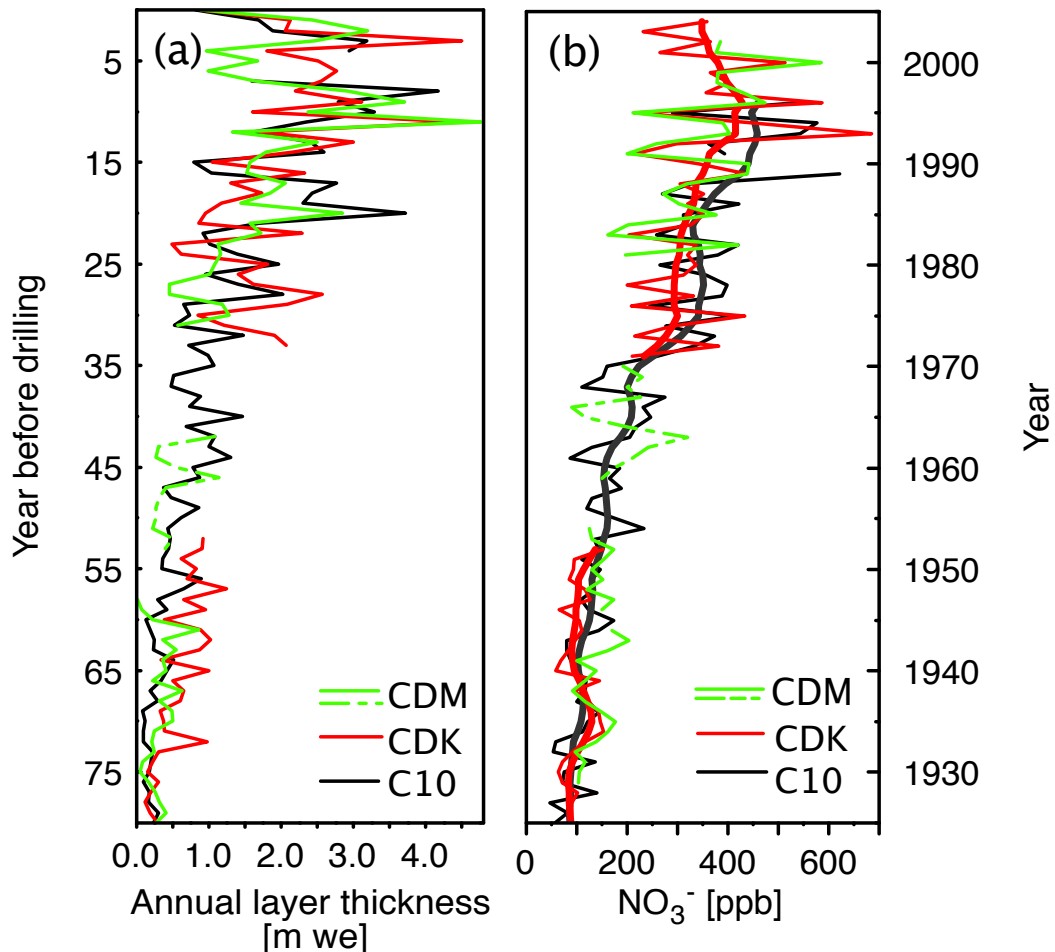

**Figure 5: (a)** Annual layer thickness of C10 (adapted from Preunkert et al. 2000), CDK (adapted from Legrand et al., 2013) and CDM. For CDM, the annual layer thickness is estimated via the ammonium stratigraphy back to 1980 and via the nitrate (and ammonium) stratigraphy further back in time (Sect. 3.1.3). **(b)** comparison of nitrate summer half-year means of C10 (adapted from Preunkert et al., 2003), CDK (adapted from Legrand et al., 2013) and CDM. The thick solid lines for C10 and CDK refer to the smoothed profiles (single spectrum analysis, see Legrand et al., 2013). CDM depth intervals for which the dating is uncertain (Sect. 3.1.3), are marked with dashed lines.

The inset in Fig. 4 (top panel) shows the $^3$H profile measured on CDM ice core samples. Ages were assigned according to the expected $^3$H concentrations based on comparison with values obtained in a high-resolution ice core from Fiescherhorn, Switzerland (Schotterer et al. 1998). Strikingly, we find not one but two distinct peaks of 183.1 ± 9.7 TU and 162.7 ± 8.8 TU (one tritium unit, TU, being equal to 0.12 Bq L$^{-1}$) less than 6 m apart in the depth intervals 87.29 to 88.2 m and 93.29 to 94.1 m depth, respectively. Both maxima are close to the $^3$H peak value normally reached in 1963. Note that analytical or sample handling errors can be excluded since this anomaly has been confirmed by independent measurements made on different

aliquots. Measurements on drill chips of the ice core from 87–88 m depth made at IUP indeed revealed 212 TU compared to 183.1 TU measured at CEP.

Based on the undisturbed depth-age scale of core C10 drilled 18 years before CDM, the 1963 maximum would be expected at about 105 m depth in CDM (49 years before the 2012 drilling year). Below 96.3 m depth, the $^3$H profile of CDM shows low values suggesting the ice is older than 1954. Relative to the disturbed depth-age relation in CDK, drilled 8 years before CDM, a 49-year-old ice layer should be located at 92 m depth, i.e., close to the deeper $^3$H peak observed in CDM. We therefore assume at this stage that the $^3$H peak observed between 93.29 and 94.1 m depth in CDM ice corresponds to 1963 but that this 1963 layer is part of a disturbed stratigraphy both in CDM and CDK. In addition, the equally high $^3$H value at around 87.5 m depth in the CDM core may point to a doubling of the same ice section in the disturbed stratigraphy.

Following what is expected from the Fiescherhorn depositional record, a mean TU value around the 1963 peak should be 10 to 40 TU in the years 1958-1975. This is consistent with the observed value in CDM ice above and below the deeper $^3$H peak (which is around 93.5 m depth). Since the $^3$H profile is available only at coarse resolution (75 cm long samples), it is not possible to be more accurate in dating the bomb test period (1954 to ~1976). Further arguments as to whether the ice layers around the deeper $^3$H peak in the CDM core are well preserved were not conclusive. For example, if we assume that the profile is continuous between 93.3 and 96.3 m depth (i.e., 1954 as indicated by $^3$H data) this would imply an annual layer thickness of 0.28 mwe over the 1963-1954 years, which is similar but (as expected from glacier flow) lower to what is seen in the C10 core (0.4 mwe). However, on the other hand the $NH_4^+$ and $NO_3^-$ depth stratigraphies from CDM do not agree with those of C10 for this depth interval. Because of decreasing anthropogenic emissions back in time, we consistently observed a decreasing trend in $NH_4^+$ and $NO_3^-$ concentrations with age in the C10 ice (mean summer value in 1964–1968: $NH_4^+$: 110 ppb; $NO_3^-$: 178 ppb; in 1963–1954: $NH_4^+$: 95 ppb; $NO_3^-$ 140 ppb). This feature, however, is not detected when comparing summer $NH_4^+$ and $NO_3^-$ means in CDM ice above and below the 1963 peak at 93.3 m. Over the 88 to 93.3 m depth interval in CDM ice, we observed mean summer $NH_4^+$ and $NO_3^-$ concentrations of 90 ppb and 175 ppb, respectively, which are lower than those between 93.3 and 96.3 m depth ($NH_4^+$: 116 ppb; $NO_3^-$: 193ppb; see also Fig. 4 for $NO_3^-$). Finally, annual layer counting based on the CDM nitrate or ammonium profiles suggests only 4 years instead of 9 years for the 93.3-96.3 m depth interval (i.e., from 1963 to 1954). If the shallower $^3$H peak is considered to be the true 1963 maximum, this would mean that a hiatus of 16 years exists in the CDM record between 1979 at 81 m depth and 1963 in 87.3 m depth. The depth layers between 88 and ~93.3 m depth would then correspond to years prior to 1963. This assumption is again in contradiction with the mean summer levels of $NH_4^+$ and $NO_3^-$ observed in C10 for this period (see discussion above). Another stratigraphic perturbation is therefore required to produce the deeper $^3$H maximum at 93.3 m.

In summary, our results suggest a continuous depth-age relation for the CDM core from the surface back to 1980 and for years older than ~1954 (96 m), implying a disturbed interval in between encompassing at least 25 years in this core. However, to confirm the assumption of the recovery of an undisturbed depth-age relation prior to 1954 done, further investigations of additional absolute time markers (as done in C10 ice, see Sect. 3.1) are needed in the future before the lower part of the CDM core can be used as an archive of past atmospheric changes.

### 3.2 A [210]Pb anomaly in the three records

Figure 6 reports the [210]Pb activity depth profiles of C10, CDK and CDM. A [210]Pb activity anomaly is found in all three cores in the depth interval of about 80-105 m below the surface, despite the drilling dates of the three cores differing by 10 years from C10 to CDK and another 8 years from CDK to CDM. Thus, this [210]Pb anomaly is not a signal that has been deposited at the glacier surface, as this would require the anomaly to be found at greater depth in the more recent cores. In contrast we assume that the ice affected by this [210]Pb anomaly comes from the same vertical depth interval of the firn column at or close to the crevasse wall and, thus, flows into the ice core site always at the same depth interval (roughly indicated by the two thick green lines in Fig. 3).

As we do not know the exact time this [210]Pb anomaly had been imprinted in the firn column and as the travel time from the crevasse to the drill site differs by a few years dependent on the depth of the ice, the [210]Pb profiles in Fig. 6 are not corrected for the decay occurring since the imprint of the anomaly in the firn column and its recovery in the ice core. As the travel time is a few decades and, thus, on the same order as the decay time of [210]Pb (half-life of 22.3 years), this implies that the initially imprinted [210]Pb signal at the crevasse was 2-3 time higher than displayed in Fig. 6. We stress that whereas [210]Pb was measured continuously on the C10 ice core, [210]Pb measurements on CDK and CDM were focused on the [210]Pb anomalies starting below 80 m. Therefore, only a few samples with limited depth and time coverage are available in the upper parts of CDK and CDM. However, comparing [210]Pb levels of the shorter CDM samples with the two samples integrating 2 and 4 years, and in view of the limited seasonal variation (with the exception of the outstanding hot summer of 2003) observed in Fig. 7, we assume the sample lengths and coverage is good enough to depict the order of magnitude of [210]Pb activities in the upper parts of the cores.

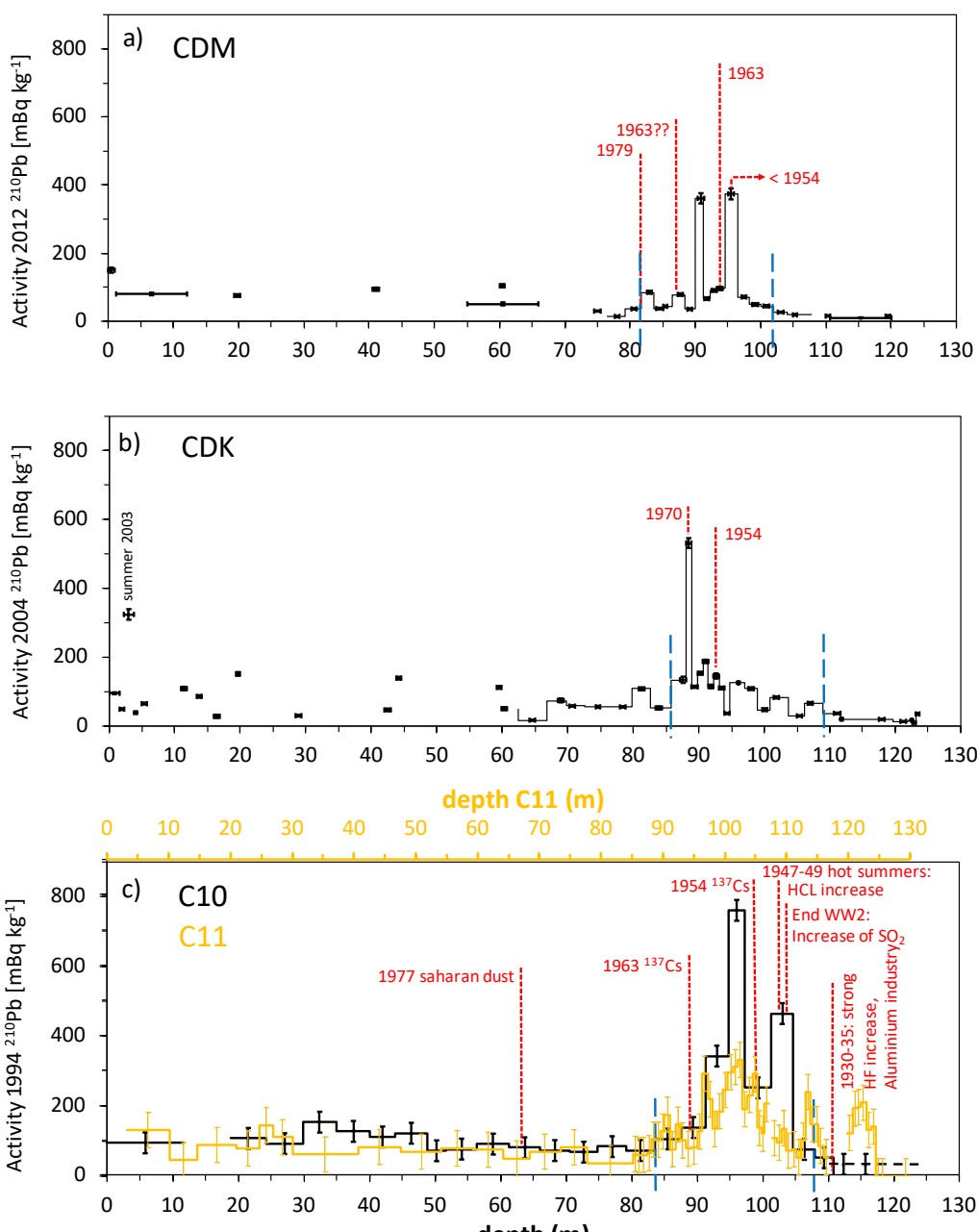

**Figure 6:** $^{210}$Pb profiles of the CDM (a), CDK (b) and C10 (c) ice cores. The $^{210}$Pb activity (only corrected for the decay of $^{210}$Pb between the drilling and analysis dates and not for the travel time from the crevasse to the drill site) is shown. For CDM (a) and CDK (b) the depths covered by the samples are plotted with thick black lines, whereas the thin lines are given to guide the eye and were used to calculate the $^{210}$Pb inventories. C10 $^{210}$Pb data (c) (lower x-axis, black) from Vincent et al. (1997) and this study are compared to those from a 140 m long ice core extracted 30 m away from C10 also in 1994 (labeled as "ice core 2" in Vincent et al., 1997 and denoted here as C11, upper x-axis in c, orange). The depth scale of C11 was aligned to achieve a match in depth in 1963 and 1954 of the respective $^{137}$Cs signals. Blue dashed vertical lines indicate the approximate boundaries of the anomaly (see also Fig. 7). When available, absolute time markers detected over the $^{210}$Pb perturbed depth zones are also reported.

In the following we discuss how this $^{210}$Pb anomaly might have been imprinted in the firn column at or close to the crevasse. $^{210}$Pb is produced through radioactive decay of the noble gas $^{222}$Rn (half-life of 3.8 days), which is an intermediate product in the normal radioactive decay chain of thorium and uranium, and emitted from the ground. $^{222}$Rn is almost entirely produced from radium in soils, in particular when granitic rocks are present. $^{222}$Rn is released from soils into the atmosphere (Dörr and Münnich, 1990; Turekian et al., 1977), and its atmospheric sink consists in its radioactive decay producing $^{210}$Pb, which becomes immediately attached to submicron aerosol particles (Whittlestone, 1990; Sanak et al., 1981). Typical atmospheric background activities at high elevation Alpine sites are 3-5 Bq m$^{-3}$ for $^{222}$Rn and typical background $^{210}$Pb activities in surface snow samples are 70-100 mB kg$^{-1}$ (Griffiths et al., 2014 and Gaeggeler et al., 2020). Accordingly, anomalously high $^{210}$Pb activities can be found over the depth interval 80-105 m in the not decay corrected records of Fig. 6 in each core which cannot be explained from $^{222}$Rn decay in the atmosphere.

Overall, three common features can be identified in the $^{210}$Pb depth profiles in each of the records. First, in the upper core sections down to 80 m, the $^{210}$Pb activities are around 100 mBq kg$^{-1}$ and of the order of magnitude of those expected from atmospheric deposition at high Alpine sites (Eichler et al., 2000; Gaeggeler et al., 2020). Looking at the continuous $^{210}$Pb record in C10 (and C11) (Fig. 6) there is a slightly decreasing trend down to 80 m depth, however, it does not account for the full expected decline by radioactive decay of about a factor of 2, which is expected from the travel time as the ice flows from the surface to the drill site and which is of similar order as the $^{210}$Pb half life. The fact that the expected decrease of $^{210}$Pb activities is smaller is not surprising since the $^{210}$Pb deposition at the glacier surface is not constant in time and space. Based on atmospheric $^{210}$Pb measurements performed at high-elevation Alpine sites (see Hammer et al. (2007) for Sonnblick at 3106 m asl, Austria and Gaeggeler et al. (1995) for Jungfraujoch station at 3450 m asl, Switzerland), it was shown that the intensity of vertical upward transport of $^{210}$Pb-rich continental boundary layer air masses strongly impacts $^{210}$Pb levels at high elevation sites. As a consequence, a strong seasonal cycle with $^{210}$Pb concentrations three to four times higher in summer than in winter is observed at high altitude Alpine sites. As expected, this also is observed in the snow deposition at CDD and shown in Fig. 7 for summer 2004 and the unusually hot summer 2003, for which an extremely enhanced upward transport was reported previously (Legrand et al., 2005). Together with the systematic decrease of the winter to summer layer thickness ratio with increasing core depth down to 75 m (from 1 to 0.6) at the drill site (see Sect. 3 and Preunkert et al., 2000), this pronounced $^{210}$Pb seasonality at least partly counteracts the expected $^{210}$Pb decrease from radioactive decay.

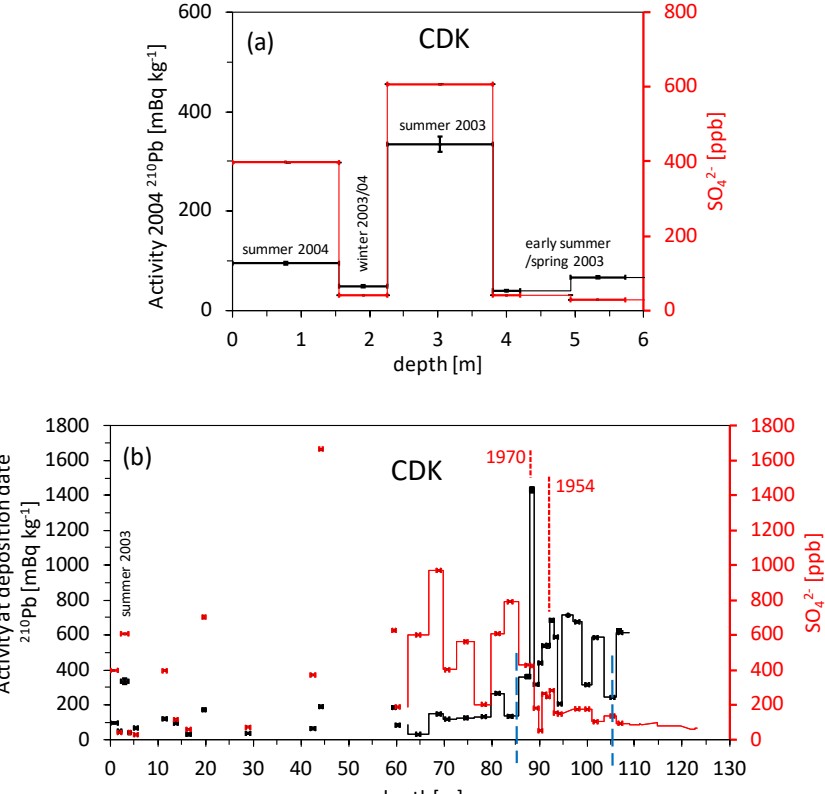

**Figure 7: ²¹⁰Pb profiles of the CDK ice cores (left y-axis, black) together with corresponding SO₄²⁻ concentrations (right y-axis, red). for the top 6 m (a) and the entire core (b). Note that in (b) the CDK ²¹⁰Pb record is tentatively corrected versus the respective snow deposition dates of the ice layers based on the CDK ice core chronology (see also Fig. 4 and Sect. 3.2). To avoid a potential overestimation of ²¹⁰Pb activities between 85 and 92 m due to the dating uncertainty, the recent date limits of the uncertainties were assigned to the samples. A thin line is given to guide the eye, whereas the depths covered by the samples are assigned with thick lines. The blue dashed vertical lines in (b) indicate the approximate boundaries of the anomaly.**

Second, there is a well-marked anomaly characterized by $^{210}$Pb enhancements (including $^{210}$Pb peaks up to 8 times higher than the values found in the shallower parts of the three cores despite increasing decay time with depth) (Fig. 6). The anomaly extends over the same depth interval in all cores (~83 to 108 m depth (i.e., ~26 to 54 years) in C10, ~85 to 108 m (i.e., ~32 to 70 years) in CDK, and ~82 to 102 m (i.e., ~33 to more than 58 years) in CDM ice. The $^{210}$Pb anomaly observed in CDK and CDM, however, is less pronounced than in C10. Since winter to summer snow ratios lie consistently between 0.5 and 0.6 in the C10 core in the depth interval of the $^{210}$Pb anomaly (Preunkert et al., 2000), these increases in $^{210}$Pb cannot be attributed to changes in seasonal snow deposition.

The starting depths of the CDK and CDM $^{210}$Pb anomalies correspond to the 1970s, for which $^{210}$Pb enhancements have been reported at other ice core sites (Eichler et al.,2000) and attributed to an enhanced vertical transport related to the temporal maximum of atmospheric sulfate aerosol acting as transport vehicle. To check whether these atmospheric conditions also could

be responsible for the enhancement seen in CDK and CDM, we report exemplarily the CDK $^{210}$Pb activity, decay corrected for its respective deposition date together with the corresponding sulfate concentration in Fig. 7. As mentioned above, a strong seasonality was detected in the uppermost part of the CDK core for a few years where $^{210}$Pb samples are available in seasonal resolution (Fig. 7a). Due to the convective co-transport of sulfate and $^{210}$Pb from the low altitude boundary layer in summer, a covariation of sulfate and $^{210}$Pb is expected, which can be seen clearly in the seasonal variation (Fig. 7). Accordingly, if caused by intensified vertical transport into the free troposphere, we would expect also higher sulfate concentrations in the depth interval, where we find the $^{210}$Pb anomalies. However, as seen in Fig. 7 the interval of highest sulfate peaks is clearly offset from the decay corrected $^{210}$Pb anomaly and sulfate concentrations strongly decrease at the depth where decay corrected $^{210}$Pb increases. Thus, the mechanism proposed by Eichler et al. (2000) cannot be invoked in this part of the CDD core. For CDM (not shown) a similar picture appears.

Third, below the anomaly, a $^{210}$Pb decrease is observed in the (not decay corrected) $^{210}$Pb data in Fig. 6 as would be expected from the increasing age, hence decay, with depth. It is worth noting that, especially in the case of the CDM and CDK cores, $^{210}$Pb activity (after blank correction) is above detection limits even in the bottommost core sections, while in C10 levels are below the detection limit. Since the age of the bottom core sections at CDD, however, exceeds several half-lives of $^{210}$Pb (as for example indicated by radiocarbon dating for CDK (Preunkert et al., 2019b)) a near zero $^{210}$Pb activity is expected if the $^{210}$Pb were only of atmospheric origin.

## 4. Discussion of upstream crevasse impact on ice core records

As indicated above, we attribute the $^{210}$Pb perturbations found at the drill site to the granite bedrock at CDD in combination with the presence of the crevasse upstream of the drill site that resulted in elevated $^{222}$Rn levels in the crevasse. In line with this assumption, Pourchet et al. (2000) conducted measurements of $^{222}$Rn in snow above a crevasse at the Mont Blanc summit, revealing unexpectedly high values of as much as 145,000 Bq m$^{-3}$, and free atmospheric background values of a few tens of Bq m$^{-3}$ at this elevation. Accordingly, the authors suggested the existence of convective $^{222}$Rn transport in the crevasse from the underlying fractured granitic bedrock and diffusion into the firn.

Since the $^{210}$Pb anomalies are located at similar depths in C10, CDK and CDM cores (Sect. 3.2), and start in the three cores ~30 years before the drilling year, we assume that the $^{210}$Pb perturbations originate from the same area upstream where the crevasse reached bedrock (at least at some time in the past) which allowed to imprint an elevated $^{210}$Pb signal in the firn close to the crevasse. Furthermore, since the $^{210}$Pb anomalies are restricted to a specific depth zone in the cores, we assume that exchange of the gaseous $^{222}$Rn with the atmosphere is restricted or eliminated at the top by the presence of a snow-bridge containing low permeability summer ice layers as have been observed to occur regularly at the site (Preunkert et al., 2000). Above the firn-ice transition of the glacier and below the snow-bridge, increased radiogenic Rn levels can then enter the firn surrounding the crevasse closed off by the snow bridge at the top.

The impact of the upstream crevasse on the depth-age relation of the ice cores changed, however, between the C10 core drilled in 1994 and the CDM and CDK cores drilled after 2000. Whereas for C10 an excellent agreement between annual layer counting and independent absolute time markers was also found over the depth interval where $^{210}$Pb was influenced by the crevasse (Fig. 6), this is not the case for the CDK and CDM ice cores. In the latter two cores, the $^{210}$Pb anomaly comprises the layers for which the depth-age relation was found to be disturbed (see Sect. 3 and Fig. 4). Furthermore, the CDK and CDM $^{210}$Pb anomaly inventories (Fig. 6) are 4 times lower than in C10, which cannot be explained by analytical differences of the measurement methods (see Sect. 2.2). We stress that the ice originating at the crevasse had essentially the same travel time from the crevasse to the drill site for a given depth in case of all three cores. Thus, the $^{210}$Pb anomaly imprinted in the firn adjacent to the crevasse was imprinted in different absolute years, as the cores were drilled 10 and 8 years apart. Accordingly, the $^{222}$Rn concentration in the crevasse leading to the $^{210}$Pb anomaly must not have been the same for all three cores, either by a change in the bedrock exposure at the bottom or by an opening of the crevasse at the top, which would have mixed the crevasse air with low $^{222}$Rn atmospheric air.

The spatial variability of the $^{210}$Pb anomaly inventory at the CDD site can be estimated by examining the $^{210}$Pb inventory of the 140 m long C11 ice core also extracted in 1994 almost at the low point of the Col du Dome saddle ~30 m south-east from C10 (Vincent et al., 1997 and see Fig. 6). This core revealed a $^{210}$Pb anomaly inventory of 80% of that in C10. Hence, although this 1994 core does not have exactly the same upstream ice flow characteristics as the 3 cores examined in this study, this difference is small compared to the difference seen between C10 on one side and CDK and CDM on the other. Again, this suggests that the C10 and C11 core saw similar $^{222}$Rn concentrations in the crevasse in the year defined by the drilling date (1994) minus the travel time of the ice from the crevasse to the drill site. In contrast, this point in time was different for CDK and CDM and the respective $^{222}$Rn concentration could have been different.

A bedrock reaching crevasse (see Fig. 3) capped by snow and ice layers at the top which is situated within the upstream flowline would lead to a continuous enrichment of $^{222}$Rn and $^{210}$Pb within the air volume of the crevasse itself as long as the bedrock is exposed, and by penetration of $^{222}$Rn also of the firn surrounding the open part of the crevasse (see the shaded areas in Fig. 3b and c). As mentioned above, the $^{222}$Rn derived from the bedrock needs to be sealed off from the atmosphere by the presence of an impermeable snow-bridge at the top to allow for high $^{222}$Rn levels in the crevasse (see dotted area in Fig. 3b and c). Below the firn-ice transition, no elevated $^{210}$Pb activities are expected as the diffusion of $^{222}$Rn into the adjacent solid ice is too slow. Tracing back the arrival depths of the $^{210}$Pb disturbance at the drill site, model calculations made by Vincent et al. (1997) and Gilbert et al. (2014) (Fig. 3a, b and c) suggest that the origin of the $^{210}$Pb anomaly should lie more than 100 m upstream of the drill site in good agreement with visual observations of the crevasse obtained via aerial and ground-based photos (see Fig. 2) and in a depth interval of 20 to 50 m below the surface (thick green lines in Fig. 3). This implies that the firn ice transition at the crevasse must lie at a depth of ~50 m (i.e. ~20 years as indicated in Fig, 2), comparable to the firn-ice transition depth of 50-55 m at the drill site. In contrast the observed firn ice transition at the summit of Dome de Gouter is located at ~25 m depth (i.e., ~100 years, Rehfeld, 2009) in line with the much lower net accumulation rates at that location.

Another question pertains whether the increased $^{222}$Rn concentrations in the firn can sufficiently affect the $^{210}$Pb activities of the adjacent firn to quantitatively explain the observed $^{210}$Pb anomalies. Concerning the entrainment of the bedrock derived $^{222}$Rn into the firn, we assume that air in the crevasse underneath the snow bridge is well mixed with respect to $^{222}$Rn and that the entrainment into the firn above the firn-ice transition is controlled by molecular diffusion. Assuming typical diffusivities D of gases in firn on the order of 0.1-1 $10^{-5}$ m$^2$ s$^{-1}$ (Birner et al., 2018), which decrease with firn density, we can estimate the entrainment length by the diffusion length $\lambda=(Dt)^{0.5}$ where t is the duration of the diffusive transport. The latter is limited by the decay time of $^{222}$Rn of 3.8 days, which controls how deep the $^{222}$Rn can enter the firn. Hence, typical entrainment lengths of 0.5-2 meters are possible during one $^{222}$Rn half live and 2 to 6 m during ten $^{222}$Rn half lives. Over this typical entrainment length, the firn will be loaded with additional $^{210}$Pb from the $^{222}$Rn decay. Compared to these entrainment velocities (a few meters in a few weeks), we can neglect the horizontal ice flow of the firn of only 4 to 5 m yr$^{-1}$.

Pourchet et al. (2000) observed mean $^{222}$Rn activities of ~10,000 up to nearly 150,000 Bq m$^{-3}$ in the firn of Mont Blanc (2 km from Dome de Gouter with the same rock mineralogy) and 0.7 Bq kg$^{-1}$ of $^{210}$Pb on average in a depth of 0.5 m in the annual snow/firn layer of the measurement year, both of which are orders of magnitude higher than typical background values in air or snow. They attributed these high levels to Rn emanation from a nearby crevasse. Vincent reported as much as 28 Bq kg$^{-1}$ in a firn/ice core at the summit of Dome de Gouter which had contact with a subsurface crevasse. Since snow accumulation and ice flow velocities at the Dome de Gouter are lower than at the Mont Blanc summit, we assume that the firn air of the core drilled at Dome de Gouter was in much longer contact with the crevasse than the surface snow layer at Mont Blanc summit.

In the following we make a rough estimate whether such $^{222}$Rn activities are sufficient to explain the $^{210}$Pb anomaly in our cores. To keep the estimation simple, we assume a temporal constant $^{222}$Rn activity in the crevasse of 50,000 Bq m$^{-3}$, which lies in the typical range observed by Pourchet et al 2000 and is equivalent to a $^{222}$Rn number concentration of 2.4 $10^{10}$ m$^{-3}$ in the crevasse. As the half life of $^{210}$Pb is much longer than that of $^{222}$Rn we can assume that the amount of $^{210}$Pb loaded into the firn is controlled primarily by the total $^{222}$Rn entering the firn. We assume that $^{222}$Rn loads the adjacent firn by diffusion and that after ten $^{222}$Rn half lives the radiogenic $^{222}$Rn entering the firn has essentially decayed to zero limiting its entrainment length to a few meters (see diffusion length discussion above). As a first order estimate we assume a linear concentration gradient between the $^{222}$Rn concentration in the crevasse and zero radiogenic $^{222}$Rn in a distance of a diffusion length after ten $^{222}$Rn half lives. We acknowledge that the true concentration gradient is not linear and that some $^{222}$Rn atoms will enter deeper into the firn than this diffusion length, but to obtain the order of magnitude loading of the firn with $^{210}$Pb this back-of-the-envelope calculation seems justified. Using the $^{222}$Rn concentration in the crevasse as a measure of the concentration gradient driving the diffusive flux, this leads to a diffusive $^{222}$Rn atom flux into the firn of 12,000 and 40,000 m$^{-2}$ s$^{-1}$ for a firn diffusivity of 0.1 $10^{-5}$ m$^{-2}$ s$^{-1}$ and 1 $10^{-5}$ m$^{-2}$ s$^{-1}$, respectively. At an exposure length of the bedrock of one year and a firn density of 500 kg m$^{-3}$, this would result in a loading of the firn by approximately 800 to 2,500 $10^6$ $^{210}$Pb atoms kg$^{-1}$, equivalent to an initial $^{210}$Pb activity on the order of 800 to 2500 mBq kg$^{-1}$. Given that the ice flows within 1 to 2 $^{210}$Pb half lives from the crevasse to the ice core drill site, these numbers are of the same order of magnitude as the activities measured in the firn core. We note that an exposure time of the crevasse of one year may be at the upper limit of what is possible (given the stationary position of the crevasse which requires healing of the crevasse before ice flow has moved its position significantly) and an exposure time of

only one month may not be sufficient to explain the measured activities. On the other hand, we only based our estimate on diffusive entrainment of $^{222}$Rn into the firn. If there also are pressure differences between the crevasse air and the firn air (for example induced by synoptic pressure variations at the surface) this would also lead to an advective flux into the firn which may increase the initial $^{210}$Pb activity after $^{222}$Rn loading of the firn. In summary, while a more precise estimate would require stringent firn transport modeling in and around the crevasse which is beyond the scope of this paper, the overall order of magnitude of the $^{210}$Pb anomaly can be explained by our simple estimate.

Although speculative, we assume that the upstream crevasse of Fig. 2 and 3 already existed earlier in the 1970s and was capped at the top by the snow bridge. If the crevasse (1) did not intersect the upstream flow line at the time the $^{210}$Pb anomaly was imprinted in the firn reaching C10 from the crevasse in 1994 (implying that the flow line was close enough to the crevasse for $^{222}$Rn to diffuse to the flow line) or (2) did intersect the catchment area of the C10 drill site at that time but was so narrow that the chronology of the C10 ice core was not significantly disturbed before closing again, this could explain the occurrence of the $^{210}$Pb anomaly observed in C10 together with an undisturbed depth-age relation. In contrast, the crevasse must have extended into the upstream flow line of cores CDK and CDM at a later date, thus not only elevating the $^{210}$Pb levels there but also disturbing their chronologies. Assuming that the crevasse was wider at that time, the appearance of winter snow enriched layers and discontinuities (i.e. lacking and/or doubling of ice layers) in the depth-age relationships in CDK and CDM drilled 10 and 18 years later than C10 could be explained by a collapse of the snow bridge partly infilling the crevasse or by closing of the crevasse void by glacier flow from upstream. Such filling with ice from glacier flow would not necessarily lead to the same stratigraphy, hence chronology, of the ice of the downstream wall and the inflowing ice. If the crevasse is empty, isochrones could end up at a greater depth than on the downstream crevasse wall (pink arrow in Fig. 3). This would result in a layer doubling at the drill site in the core, as suggested in CDM. If the bottom of the crevasse is filled with ice from the collapsed bridge before healing, inflowing isochrones could end up at a lower depth than on the downstream wall (orange arrow in Fig. 3). This would result in missing layers at the drill site in the core, as suggested in CDK.

We stress that although a $^{210}$Pb anomaly is visible in all three cores, the $^{210}$Pb inventories of CDM and CDK amount to only ~25% of the C10 inventory. Thus, the initial $^{210}$Pb loading of the firn was likely lower for CDK and CDM. This could be explained by either a shorter or reduced opening of the crevasse at the bottom at the time the firn was loaded with $^{210}$Pb for CDK and CDM or a (temporary?) opening of the crevasse to the atmosphere. A partial opening of the crevasse to the atmosphere would allow the bedrock-derived $^{222}$Rn in the crevasse to mix with the much lower atmospheric $^{222}$Rn concentrations (Pourchet et al., 2000). This would have led to a strong reduction of additional $^{222}$Rn accumulation and $^{210}$Pb production in the crevasse and in the snow and firn around the crevasse, starting from the moment of the opening to the atmosphere. In addition, disturbed isochrones also could lead potentially to decreased $^{210}$Pb inventories since ice layers from the upstream side of the crevasse with enriched $^{210}$Pb activities could be missing, as suggested for CDK but not CDM.

## 5. Summary and conclusion

Combining existing and new chemical depth profiles, bomb test time markers, and the $^{210}$Pb depth profiles of three ice cores

extracted at the same drill CDD site in 1994, 2004 and 2012, allowed us for the first time to highlight changes over time in the depth-age characteristics at an alpine drill site. Because of the granitic bedrock prevailing at the site, the imprint of a crevasse located upstream of the drill site is visible in all three ice cores as a distinct anomaly in their $^{210}$Pb profiles extending over just a few meters in depth and with $^{210}$Pb concentrations elevated by up to a factor of 10. Whereas the depth-age relationship of the C10 ice core drilled in 1994 does not appear to be disturbed by the crevasse in the upstream region, this is not the case for the CDK and CDM ice cores drilled in 2004 and 2012. For CDK and CDM, the depth-age relationships were found to be disturbed in ice layers deposited ~30 year before drilling and over a period of around 16 years in CDK and at least 25 years in CDM. We attribute this to a lateral extension of the crevasse over time into the upstream flowline of our drill site. This finding is consistent with long-term glaciological observations that show significant glacier thickness changes in the area surrounding the upstream crevasse.

Although at this stage we can provide only a qualitative explanation for the recently observed stratigraphic discontinuities, our work points towards the need for careful examination of depth-age relationships, when using ice cores from this CDD drill site to reconstruct past atmospheric conditions. More generally, since crevasses are often present on steep, non-polar glaciers, such disturbances in the depth-age relation, as observed at CDD, could also appear at other non-polar ice core drill sites but may be undetected. Particularly, when the bedrock is not granitic and does not allow to use the $^{210}$Pb imprint caused by $^{222}$Rn emanation from the bedrock, when few or no absolute time markers are available, and/or when only one core is collected from the site. To identify such depth-age problems, in addition to commonly used annual layer counting, an extended use of absolute time markers including bomb horizons through $^{3}$H, $^{137}$Cs, or $^{239}$Pu (Arienzo et al., 2016), $^{39}$Ar (Feng et al., 2019), large Saharan dust events or volcanoes (e.g., Plunkett et al., 2022) is mandatory. Furthermore, at other non-polar sites where the net snow accumulation is far lower than at CDD (i.e., with ice as old as several thousands of years located well above the bedrock), additional tools like $^{14}$C measurements (Jenk et al., 2006 and 2009; Hoffmann et al., 2018) should be applied.

**Data availability**

Ice core data are available at NCEI (National Centers for Environmental Information) data base (https://www.ncei.noaa.gov/access/paleo-search/study/38020 ).

**Author contribution**

SP, PB and ML performed research and wrote the original manuscript. SP, HF, and JRM revised the manuscript. HF, TE, RP, LZ, AW, JRM analyzed ice samples and data, and commented the original manuscript. AG did model calculations and commented the original manuscript.

## Acknowledgements

The ice core drilling operations at CDD were supported by the European Community via ENV4-CT97 (ALPCLIM) contract, the EU CARBOSOL project (contract EVK2 CT2001-00113), and the Region Rhône-Alpes. The LEFE-CHAT (CNRS)

program entitled "Evolution séculaire de la charge et composition de l'aérosol organique au dessus de l'Europe (ESCCARGO)" provided funding for analysis in France with the support of ADEME (Agence de l'Environnement et de la Maîtrise de l'Energie). NSF Grant 1925417 to J. R. McConnell provided partial support for the analyses and interpretation at DRI. CEP acknowledges the longer-term financial support of ice core research by the Swiss National Science Foundation. P.Bohleber gratefully acknowledges funding by the Austrian Science Fund (FWF) I 5246-N. The authors thank all colleagues who

participated in the drilling campaigns at CDD in 1994, 2004 and 2012, and the laboratory analyses at IUP, CEP and DRI. We also would like to thank Alison Criscitiello and three anonymous reviewers and the editor Kristin Poinar, for their thorough reviews and helpful suggestions. S.P., P.B., H.F., L.Z. und A.W. thank their late teacher Dietmar Wagenbach for his inspiring ideas on the impact of glaciological and atmospheric processes on ice core records.

**Competing interests**

The authors declare that they have no conflict of interest.

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
