# Peer review of "Impact of subsurface crevassing on the depth-age relationship of highalpine ice cores extracted at Col du Dôme between 1994 and 2012"

_The Cryosphere, 2022_

## Author Comment (AC2)

**Response to reviewer**

We would like to thank referee#2 for the detailed review of our manuscript. The comments made by the referee were appreciated and helped improve the manuscript. Please find in the attached pdf our responses to the comments (in blue) and our proposed changes to a potential revised manuscript (in blue and in italic):

**RC2: 'Comment on tc-2022-259', Anonymous Referee #2, 22 Mar 2023 reply**

The manuscript presents nitrate records obtained from three different ice cores collected at nearly the same location on Col du Dome, Mont Blanc in the years in 1994, 2004 and 2012. Using the records of nitrate and the radionuclides 3H and 210Pb together with a 3D ice flow model the authors argue that there are discontinuities in the depth-age relation of the ice cores drilled in 2004 and 2012, which were caused by the presence of an upstream crevasse. This is an interesting hypothesis. Although it is common knowledge that areas with upstream crevasses should be avoided as ice core drilling sites, it could be valuable to demonstrate what the effects of such a crevasse are. However, I find the argumentation rather speculative and not well supported by the data, which are often inconclusive. Further, I miss in some part scientific rigorousness as outlined below. Considering all my concerns as outlined below, this manuscript requires major revisions.

**Major comments**

210Pb data presented in Fig. 4 have very different time resolution. It is scientifically not sound to compare such data. For instance, the peak at 1970 in CDK would disappear if the same averaging period as in the upper part of the record would be applied. This peak is most likely due to the strong 210Pb seasonality (Eichler et al., 2000). When using the same temporal averaging the postulated anomaly in the 1960s and 1970s in the CDK and CDM cores will be much smaller and may be due to an increased input of 210Pb in the 1970s. Such an increase has been observed already at other glaciers in the Alps, e.g. at Silvretta and Adamello (Festi et al., 2021), at Colle Gnifetti (Gäggeler et al., 1983) and at Grenzgletscher (Eichler et al., 2000) and was attributed to enhanced vertical transport related to the maximum in sulphate aerosol particles acting as transport vehicles (Eichler et al., 2000).

We regret the ambiguity in the presentation of the original version of Fig. 4. Since we were focused on the depth range corresponding to the depth-age anomaly which was sampled in relative high resolution, we only sampled the upper part for of CDK and CDM at lower frequency and interpolated the measurements. This will be changed in the revised version (showing raw data). Samples of CDK and CDM now are shown in Fig, 4 as measured and the depth intervals of the individual measurements (typically 0.7 to 1 m) are assigned clearly. In addition, as requested, we discuss whether or not the re-increase of 210Pb in the 1970s in CDK and CDM might be of atmospheric origin.

Figure 4: 210Pb profiles of the three CDD ice cores. The decay-corrected 210Pb activity is shown using the drilling year of the respective ice cores as reference. *For CDK and CDM the depths covered by the samples are plotted with thick black lines, whereas the thin integrating line is given to guide the eye and was used to calculate the 210Pb inventories. Blue dashed vertical lines indicate the approximate boundaries of the anomaly. Where available, absolute time markers detected over the 210Pb perturbed depth zones are also reported. C10 data are from Vincent et al. (1997) and this study.*

The discussion whether or not the re-increase of 210Pb in the 1970s in CDK and CDM might be of atmospheric origin was revised as follows in Section 3.2:

" As a consequence, a strong seasonal cycle with 210Pb concentrations three to four times higher in summer than in winter is observed at high altitude Alpine sites. As expected, this also is observed in the snow deposition at CDD and shown in Fig S1 of the Supplement for summer 2004 and the outstanding hot summer 2003, for which an extremely enhanced upward transport was already reported previously (Legrand et al., 2005). Whereas in 2004 a summer to winter 210Pb ratio of 2 was found, this ratio reached a factor of 7 in 2003. Together with the systematic decrease of the winter to summer layer thickness ratio with increasing core depth at the drill site (see Section 3 and Preunkert et al., 2000), this pronounced 210Pb seasonality counteracts the expected 210Pb decrease from radioactive decay.

Second, a well-marked anomaly characterized by 210Pb enhancements (including 210Pb peaks up to 10 times higher 210Pb than expected from atmospheric deposition) is observed in the three cores. The anomaly extends from ~83 to 108 m depth (i.e., ~26 to 54 years) in C10, ~85 to 108 m (i.e., ~32 to 70 years) in CDK, and ~82 to 102 m (i.e., ~33 to more than 58 years) in CDM ice. The 210Pb re-increase observed in CDK and CDM, however, is less pronounced than in C10. In addition, the starting depths of the CDK and CDM 210Pb re-increases correspond to the 1970s, for which 210Pb enhancements have been reported at other ice core sites (Eichler et al.,2000) and attributed to an enhanced vertical transport related to the temporal maximum of atmospheric sulfate aerosol acting as transport vehicle. To check whether these atmospheric conditions also could be responsible for the enhancement seen in CDK and CDM, we report exemplarily the CDK 210Pb activity, corrected for its respective deposition date together with the corresponding sulfate concentration in Fig. S1 of the Supplement. As mentioned above, a strong seasonality was detected in the uppermost part of the CDK core for a few years where 210Pb samples are available in seasonal resolution (Fig. S1a of the Supplement). If atmospherically derived, mean 210Pb concentrations of ice layers from 60 to 85 m depth (i.e., from 1988 to 1972), i.e., in the period for which the sulfate aerosol maximum was observed at CDD (Preunkert et al., 2001), would correspond to around  $130 \pm 60 \text{ mB kg}^{-1}$  of 210Pb in freshly deposited snow, which is comparable to the atmospherically derived 210Pb further upward in the core. However, from 85 to 108 m depth, this connection between sulfate levels and 210Pb activity no longer holds. Whereas sulfate concentrations strongly decrease, 210Pb at the time of deposition (decay-corrected) would be strongly enhanced (mean of 600 mBq kg-1) and far above what is expected from atmospheric 210Pb contributions. Thus, the mechanism proposed by Eichler et al. (2000) cannot be invoked in this part of the CDD core. For CDM (not shown) a similar picture appears. While from 80 to 90 m surface decay-corrected 210Pb (160  $\pm$  70 mBg kg-1) would not have been significantly enhanced compared to the atmospherically derived 210Pb concentrations seen further up in the CDM core, this is not the case between 90 and 103 m depth. As for CDK, mean values at the time of deposition would have been around 650 mBq kg-1 and thus far too high to what would be expected from atmospheric transport.

Third, below the anomaly, a decrease ....

Revised Fig. S1 of the Supplement now is as follows:

---

## Author Response (AR1)

Dear editor,
Thanks for your comments. We followed all your suggestions. Please find in the following our respective responses to the comments (in blue) and our proposed changes for a potential revised manuscript (in blue and in italic).
Sincerely
Susanne Preunkert

**Response to the editor comments**

Editor:
Your manuscript has now received two detailed peer reviews that highlight the strengths and shortcomings of the manuscript. The reviews were constructive and supportive of your work. They each identified specific issues that would need to be addressed in a revised version of the manuscript in order for potential future publication in The Cryosphere. Thank you for your Author Comment responses to these reviews. I find your responses, proposed changes, and already modified drafts of figures to be promising. As you no doubt noted while preparing your responses, both reviewers (as well as myself) felt muddled in finding the story and sequence of events that the crevasse went through in order to generate the situations observed in the three boreholes. I believe that clarifying this will be an important component of your revision.

I have a number of comments, in no particular order, that originated as I read the reviews, responses, and the manuscript again. Please take these, as well as your Author Comment responses, into account in your revision.

- The proposed revisions to Figure 5 are helpful. I would benefit from seeing the approximate time (e.g. 1995-2000, or whatever you estimate) marked on each schematic (b-c) or noted in the caption.

Thanks for this comment, these dates are now added in the figure caption (see below).

- I would reiterate both reviewers' requests that the sequence of events that the crevasse went through be reported more clearly. This should not involve parenthetical ties to the data, as you currently propose in the revision. Rather, give the narrative story that you propose for the crevasse up front, and then in subsequent paragraphs, explain how the data support that story uniquely. One way to achieve this would be to move Figure 5 to appear much earlier in the manuscript, perhaps as Figure 2. This would give your readers an early encounter of what you propose and would allow their understanding to develop as they read your data. My understanding of your manuscript is that the crevasse story is the primary takeaway, and the new datasets (210Pb, nitrate, etc.) are also new scientific contributions, but they are secondary and are in support of the crevasse idea. Thus, they should appear as later figures than the crevasse figure.

Thanks for your comment. We understand your concern and the concern of the reviewers about this point, which we have tried to address in what we hope is a suitable compromise. We followed your idea to present the crevasse in more details at the beginning of the manuscript. To do so, we introduced the crevasse clearly in Section 2 and moved Figure 5 (now Figure 2) just after Figure 1. With that, the reader will indeed be aware of the crevasse location with respect to the drill site, the fact that it changed over time and the effect of 222Rn and 210Pb accumulation in the crevasse under sealed conditions. However, we believe that we cannot discuss further the evolution of crevasse before adequately discussing all data presented in Section 3. Only through the data presented here the magnitude and temporal extent of this effect becomes evident, which is the basis for discussion the evolution of the crevasse. Therefore, we think that the detailed discussion about the crevasse should stay in Section 4, constituting the synthesis part of the manuscript. The proposed respective paragraph added to Section 2 would read as following in the revised manuscript.

*"...Ice flow, firn compaction and thermal regime have been modeled in three dimensions by Gilbert et al. (2014), allowing particle back-tracking and flow-based estimation of the depth-age relationship for the drilling site.*
*Recent visual observations made on the CDD glacier attest to the presence of crevasse(s) upstream the CDD drill sites (Fig. 1). Comparing photos taken in 2012 (Fig. 1a) and in 1999 (Fig.1b), shows an enlargement and horizontal propagation of the crevasse to the east from 1999 to 2012. Whereas in 2012, the crevasse is visible clearly as a snow-covered depression on the surface slope, the crevasse appeared to be limited to the southwestern flank of the drill site catchment area in 1999. Following Fig. 1, the crevasse is situated ~100-150 m upstream of the drill site of C10, CDK, and CDM. Figure 2a shows the CDD glacier thickness changes between 1993 and 2017 overlayed with the modelled flow line indicating the calculated arrival depths at the drill site of C10, CDK, and CDM (Gilbert et al., 2014). Crevasses open and close constantly during their lifecycle (Colgan et al., 2016). Fig. 2b and c represent vertical cross sections along the modelled flow line of Fig. 2a overlayed by sketches of the upstream crevasse visible in Fig.1, in two temporal states of the crevasse from the 1960ties to the 1980ties, as concluded in Section 4 on the basis of C10, CDK and CDM ice core data presented in Section 3.*
*.."*

[Figure]

**Figure 2: (a)** Thickness changes between 1993 and 2017. The contour lines of surface topography correspond to the 1993 surface (adapted from Vincent et al., 2020) overlain by a modelled flow line (color scale on top) which reports the calculated arrival depth at the drill site of C10, CDK, and CDM (black star) (Gilbert et al., 2014). The crevasse location (blue line) is based on the 30th June 2004 aerial photo from IGNF (see Fig.1) **(b and c)** Schematic representation of the origin of the $^{210}$Pb anomalies found at the drill site following the ice flow model of Gilbert et al., 2014, extracted along the flow path reaching the drill site. Isochrones are marked in red, flowlines in green (see also Section 4). The grey shaded zone indicates firn, the dotted zone indicates the snow bridge over the crevasse. Concluded from ice core data of C10, CDK and CDM (see Section 3 and 4), two states of the crevasse are reported**: (b) in the years ~1965-1970 (i.e. ~ 25-30 years before the C10 drilling)** the crevasse is open to the bedrock but sealed from the atmosphere by a snow bridge. In this state $^{222}$Rn and $^{210}$Pb accumulate to reach concentrations well above atmospheric conditions in the crevasse and the surrounding firn **(c) after ~1975 and at least until ~1990 (i.e. ~ 25-30 years before the CDK and CDM drilling),** the crevasse is at least partly open to the atmosphere. In this state $^{222}$Rn and $^{210}$Pb concentrations in the crevasse and the surrounding firn are strongly reduced compared to (b). The formation of missing or doubling ice layers is indicated by the orange and pink arrows.

- It is not clear to me how the addition of two new samples to the Vincent et al. (1997) record would explain the significantly greater 210Pb activity peaks in the new C10 data. Please clarify your proposed revision in response to Reviewer 2's question about these peaks.

We apologize for the confusion related to the fact that our data differ from those in Vincent et al. (1997) but since the data were measured such a long time ago, we have forgotten that there were the differences of the two peaks in the two data sets. Sorry for that.

For your information, in the year after the study of Vincent et al. (1997) was published, a few further 210Pb spot checks measurements were made at LGGE on the C10 ice core using ice core samples instead of drilling chips this time, to confirm (or not) the enhanced 210Pb levels found in C10 and measured on the drilling chips. These additional measurements included also the C10 core depth sections for which Vincent et al. (1997) did not assign values since the high 210Pb values raised at a first-time doubt of contamination. Since the second measurements confirmed however the results of the first sample measurements, we kept the two 210Pb peaking samples (from94.8 to 97.3 m and 101.3 to 104.7 m) in the data set.

This is the main difference between the Vincent et al. (1997) and our dataset. We revised our proposed revision of this point in Section 2 and give now more details about the two samples which are included in our data but not included in the original dataset of Vincent et al. (1997).

*"Note that, the dataset from Vincent et al. (1997) was complemented by two additional samples for which $^{210}$Pb analysis and quality control were not available in 1997. Initially suspected to be contaminated, these two samples, containing 760 and 460 mBq kg$^{-1}$of $^{210}$Pb, were not included in the dataset reported by Vincent et al., 1997. Re-measurements of the respective ice core sections using samples extracted from the inner of the core confirmed however the initially measured values, hence they must be considered as valid and were included in the data set of this study.*

To avoid further confusion, we re-compared our data set directly with the data given in Fig. 5 in Vincent et al. (1997). All data points have the same $^{210}$Pb level except the data point between 97.3 to 101.3 m. This sample indicated an excess of 50 mBq kg$^{-1}$ compared to our data set probably due to the second series measurements made after 1997. To keep (except the two added samples) one unique data set in literature, this data point was aligned in our series to match now with the ones given by Vincent et al. (1997).

- The decay chain leading to 210Pb should be better described in the Introduction or Methods (perhaps Section 3.2, where 210Pb is described extensively). The proposed change, to mention this parenthetically in Section 4, would not be helpful enough. This needs to be more prominent and upfront.

Thanks for this comment, the decay chain of $^{210}$Pb is now detailed at the beginning of Section 3.2. Our proposition for the revised manuscript is:

*"..Figure 5 reports the $^{210}$Pb (half-life of 22.3 years) depth profiles of C10, CDK and CDM. $^{210}$Pb is produced through radioactive decay from the noble gas $^{222}$Rn (half-life of 3.8 days), which is an intermediate product in the normal radioactive decay chain of thorium and uranium, and emitted from the ground. $^{222}$Rn is almost entirely produced from Radium in soils, in particular when granitic rocks are present. $^{222}$Rn is released from soils into the atmosphere (Dörr and Münnich, 1990; Turekian et al., 1977), and its atmospheric sink consists in its radioactive decay producing $^{210}$Pb, which becomes immediately attached to submicron aerosol particles (Whittlestone, 1990; Sanak et al., 1981). "*
*"*

- I very much like the proposed Figure S2, which shows the 210Pb records for both C10 and C11 cores. Rather than adding a supplement, I would suggest that this simply be swapped in for panel c on Figure 4.

Ok, we swapped Figure S2 in Figure 4c (will be Figure 5c) in the revised manuscript.

[Figure]

**Figure 5: $^{210}$Pb profiles of the three CDD ice cores. The decay-corrected $^{210}$Pb activity is shown using the drilling year of the respective ice cores as reference. For CDK (a) and CDM (b) the depths covered by the samples are plotted with thick black lines, whereas the thin integrating lines are given to guide the eye and were used to calculate the $^{210}$Pb inventories. C10 $^{210}$Pb data (c) (lower x-axis, black) from Vincent et al. (1997) and this study are compared to the ones of a 140 m long ice core extracted 30 m away from C10 in 1994 (Vincent et al., 1997, denoted here as C11, upper x-axis in c, orange). The depth scale of C11 was matched to achieve an overlay of the depths in 1963 and 1954 obtained from the respective $^{137}$Cs signals. Blue dashed vertical lines indicate the approximate boundaries of the anomaly. When available, absolute time markers detected over the $^{210}$Pb perturbed depth zones are also reported.**

**Response to reviewer**

We thank referee#1 for reviewing our manuscript and we much appreciate his suggestions which helped improve the manuscript. Please find in the attached pdf our responses to the comments (in blue) and our proposed changes to a potential revised manuscript (in blue and in italic):

**RC1**: 'Comment on tc-2022-259', Anonymous Referee #1, 28 Feb 2023  reply
Review of Preunkert et al." Impact of subsurface crevassing on the depth-age relationship of high alpine ice cores extracted at Col du Dome between 1994 and 2012

Preunkert et al. compare the records of three ice cores drilled at Col du Dome, near Mont Blanc in 1994, 2004, and 2012. The age scale appears intact in the 1994 core (C10) while the age scales are disturbed in the 2004 (CDK) and 2012 (CDM) cores in the time period of the 1950s and 1960s. The dating is primarily established by annual layer interpretation of ammonia, but the disturbances are primarily identified by the complexity of the H3 and C137 records. They ascribe the disturbances to the presence of a crevasse upstream. The crevasse, which is sealed near the surface by a snow/ice bridge, allows the accumulation of Pb210 due to the granitic bedrock. I believe the primary argument is that the dated ice in the 1994 core originated when the crevasse was smaller and did not yet intersect the flow path reaching the ice core site. The 2004 and 2012 were disturbed, however, because the crevasse had enlarged and intersected the flow path.

Preunkert et al. present high quality measurements of a large variety parameters and provide a plausible explanation for the disturbed stratigraphy in the two later cores. The use of the bomb horizons to evaluate disturbances is an interesting application. The primary conclusion that care must be taken in interpreting alpine ice core timescales is well supported. The mechanisms of layer skipping and layer doubling is well established. I have a few suggestions to improve the manuscript and make the argument more convincing.

The extension of the crevasse through time should be presented in more detail. A plan view of the extension would be very helpful. The photos in Figure 1, particularly 1b, is quite poor. Given the popularity of Mt. Blanc, it seems like a long record of photographs exists to validate the hypothesis of crevasse extension. Mapping of the crevasse through time would significantly improve the plausibility of the proposed mechanism.

Thanks a lot for this comment and the good idea concerning the mapping of the crevasse over time. Unfortunately, it was not possible to find photos showing precisely that view on the Dome du Gouter and the crevasse. Given the high accumulation at the site (around 2 mwe per year, i.e. 4-6 m of snow per year at the location of the crevasse), it is not surprising that it is partly and temporarily closed and hard to see on the photo.
We checked on the web, and found many photos showing the slope which rises to the Vallot Observatory (and the photogenic ridge rising at the Mt Blanc), but hardly any from Vallot showing the Dome. We asked colleagues, and we rechecked our own collection of photos from the site, but the one that was included is the best we found from the period around the year 2000 or earlier. Therefore, we have to stay with the original photo.

A plan view of the crevasse is assigned in Figs. 1 and 5 on the basis of an aerial photo (from 2004) from the Institut national de l'information géographique et forestière (IGNF). In this database, we found one photo among many in which one could at least imagine the crevasse. In the original manuscript the line was however drawn too thin. This is changed now and in both Figs. the plan view of the crevasse is better indicated.

I found the discussion of Pb210 and Rn222 to be rather confusing. I didn't see any data on Rn222 presented and am unclear how this fits into the Pb210 and crevasse story.
As stated in the introduction of the manuscript, 222Rn (half-life of 3.8 days) is emitted from bedrock, especially from granite. 222Rn is the radioactive gaseous precursor of 210Pb (half-life of 22.3 years). Thus, 222Rn is the source of 210Pb which is produced through radioactive decay. This important relation will be emphasized in the beginning Section 4: *"Furthermore, since the $^{210}$Pb anomalies are restricted to a specific depth zone in the cores, we assume that exchange of the gaseous $^{222}$Rn (i.e., the radioactive precursor of $^{210}$Pb) with the atmosphere is restricted or eliminated at the top by the presence of a snow-bridge containing horizontal summer ice layers such as ...."*

The authors also reference Pb210 record from 30m away, but this is not shown. It would be helpful to see how this compares to the C10 record and strengthen the arguments.
A comparison of the 210Pb record of the C11 ice core drilled 30 m away with the C10 record will be shown in Fig. S2 of the Supplement:

[Figure]

*Figure S1: $^{210}$Pb profiles of the CC10 (lower x-axis, black) compared to the one of a 140 m long ice core extracted 30 m away from C10 in 1994 (Vincent et al., 1997, denoted here as C11, upper x-axis, blue). The decay-corrected $^{210}$Pb activity is shown using the drilling year of the two cores as reference. The depth scales of both cores were matched to achieve an overlay of the depths in 1963 and 1954 obtained from the respective $^{137}$Cs signals. C10 and C11data are from Vincent et al., 1997. C10 data were completed in this study.*
.

But mainly I remain unclear on why C10 is more enriched in Pb210 if the ice did not intersect the crevasse.

As mentioned above, the point is that the source of 210Pb is the noble gas 222Rn which is an intermediate product in the normal radioactive decay chain of thorium and uranium, and emitted from the ground. 222Rn (half life of 3.8 days) can diffuse in porous snow and firn material and decay to become 210Pb there (half life 22.3 years). The layers enriched in 210Pb would them become part of the ice column and be transported by ice flow. Therefore, 210Pb can be enriched without a direct intersection of the crevasse with the ice core.

We will clarify this point in in the beginning Section 4:

"*Furthermore, since the $^{210}Pb$ anomalies are restricted to a specific depth zone in the cores, we assume that exchange of the gaseous $^{222}Rn$ (i.e., the radioactive precursor of $^{210}Pb$) with the atmosphere is restricted or eliminated at the top by the presence of a snow-bridge containing horizontal summer ice layers such as ...."*

This is a complex system which necessitates temporal variations in the crevasse as well as coverage of the crevasse with a snowbridge and the firn/ice transition. A schematic showing different crevasse and firn configurations and the resulting Pb210 anomalies would be very helpful.

We fully agree about the complexity of this system. Essentially there are two states of the crevasse. One for which the crevasse is open to bedrock and sealed by a snow bridge, and a second in which it is at least partly open to the atmosphere. Whereas in the first state the 222Rn emitted from the granite in the bedrock will accumulate, diffuse into the surrounding firn and produce 210Pb in excess there (this would correspond to what is observed in C10), in the second state the excess 222Rn gas can escape from the crevasse to the atmosphere, thus 210Pb production will be strongly limited (this would correspond to what is observed in CDK and CDM). As you suggested these two states are now reported in Fig 5b and 5c.

[Figure]

[Figure]

**Figure 5: (a)** Thickness changes between 1993 and 2017. The contour lines of surface topography correspond to the 1993 surface (adapted from Vincent et al., 2020) overlain by a modelled flow line (color scale on top) which reports the calculated arrival depth at the drill site of C10, CDK, and CDM (black star) (Gilbert et al., 2014). The crevasse location (blue line) is based on the 30[th] June 2004 aerial photo from IGNF (see Fig.1) **(b and c)** Schematic representation of the origin of the [210]Pb anomalies found at the drill site following the ice flow model of Gilbert et al., 2014, extracted along the flow path reaching the drill site. Isochrones are marked in red, flowlines in green (see also Section 4). The grey shaded zone indicates firn, the dotted zone indicates the snow bridge over the crevasse. *Two states of the crevasse are reported: (b) the crevasse is open to the bedrock but sealed from the atmosphere by a snow bridge. In this state [222]Rn and [210]Pb accumulate to reach concentrations well above atmospheric conditions in the crevasse and the surrounding firn (c) the crevasse is at least partly open to the atmosphere. In this state [222]Rn and [210]Pb concentrations in the crevasse and the surrounding firn are strongly reduced compared to (b). The formation of missing or doubling ice layers is indicated by the orange and pink arrows.*

In addition we will reword the discussion of this point in Section 4 in the following way:

*"... A partial opening of the crevasse to the atmosphere would allow the bedrock-derived [222]Rn in the crevasse to mix with the much lower atmospheric [222]Rn concentrations (Pourchet et al., 2000). This would have led to a strong reduction of additional [222]Rn accumulation and [210]Pb production in the crevasse and in the snow and firn around the crevasse, starting from the moment of the opening to the atmosphere. This would explain [210]Pb inventories of 70 and 55% in CDK and CDM compared to C10, because of the radioactive decay of [210]Pb accumulated before the opening of the crevasse to the atmosphere, over 10 and 18 years, respectively......"*

A few additional minor comments and/or questions:
L266 – "reach"
ok done

Have cores been drilled on Dome de Gouter? The ice thickness may be less and the accumulation lower, but couldn't these cores provide good benchmarks to compare the records collected at Col du Dome?
There was one core drilled on Dome du Gouter, however processing of the core and the data is not finished and there are no 210Pb data available. Furthermore, it is very likely that a full seasonal cycle of snow accumulation will not be well preserved there (due to preferential wind erosion in winter) rendering more delicate the use of the chemical ice-core record for atmospheric chemistry.

Figure 2 – is there an a priori expectation for the H3 and C137 profiles that could be plotted behind the measurements?
The 3H and 137Cs signals found in Alpine glaciers are related to the atmospheric nuclear tests conducted from 1954 (the beginning of atmospheric fall-out) to 1974. It is well established that the maximum radioactivity in precipitation in the Northern hemisphere was in 1963. Among the long-lived products from these events are 137 Cs (half-life of 30.15 years), 90 Sr (28.15 years) and 3 H (12.34) years.

Considering that the information conveyed in Fig. 2 is already very dense, we decided to add this information to the text in Section 3.1:
"…and radiometric analyses aimed at detecting fallout from atmospheric thermonuclear bomb testing via $^3H$ (Legrand et al., 2013 for CDK and this study for CDM) and $^{137}Cs$ (Vincent et al., 1997) for C10, as already done for other Alpine ice cores records (e.g. Schotterer et al., 1998). Fallout from atmospheric thermonuclear bomb testing typically leads to elevated $^{137}Cs$ and $^3H$ levels from 1954 to about 1975, with maxima in 1963 if the depth-age relationship is well preserved. The $^{210}Pb$ depth …"

and in Section 3.1.1:
"The dating of the C10 core was found to be in excellent agreement with several outstanding atmospheric changes or events that occurred during the 20th century such as the $^{137}Cs$ peak caused by nuclear weapons testing fallout (Vincent et al. 1997), the well-marked increase of fluoride after 1930 ….."

Figure 2 – it would be helpful to have the annual layers marked, at least on the CDM profile
Ok this is done, for the upper part of CDM (back to 1981) which could be dated reliably.

Figure 4 – please make the y-axes the same on all plots so that the differences in magnitude – which I believe is the primary point – stands out more clearly. And please include the results from the core 30m away

Ok this is done, y-axes are changed and the core from 30m away will be reported together with C10 in Fig. S2 (see also our comment above).

Figure 5 – make the bedrock a thicker line and different color
Ok done (see above).

**Response to reviewer**

We would like to thank referee#2 for the detailed review of our manuscript. The comments made by the referee were appreciated and helped improve the manuscript. Please find in the attached pdf our responses to the comments (in blue) and our proposed changes to a potential revised manuscript (in blue and in italic):

**RC2**: 'Comment on tc-2022-259', Anonymous Referee #2, 22 Mar 2023  reply
The manuscript presents nitrate records obtained from three different ice cores collected at nearly the same location on Col du Dome, Mont Blanc in the years in 1994, 2004 and 2012. Using the records of nitrate and the radionuclides 3H and 210Pb together with a 3D ice flow model the authors argue that there are discontinuities in the depth-age relation of the ice cores drilled in 2004 and 2012, which were caused by the presence of an upstream crevasse. This is an interesting hypothesis. Although it is common knowledge that areas with upstream crevasses should be avoided as ice core drilling sites, it could be valuable to demonstrate what the effects of such a crevasse are. However, I find the argumentation rather speculative and not well supported by the data, which are often inconclusive. Further, I miss in some part scientific rigorousness as outlined below.  Considering all my concerns as outlined below, this manuscript requires major revisions.

**Major comments**

210Pb data presented in Fig. 4 have very different time resolution. It is scientifically not sound to compare such data. For instance, the peak at 1970 in CDK would disappear if the same averaging period as in the upper part of the record would be applied. This peak is most likely due to the strong 210Pb seasonality (Eichler et al., 2000). When using the same temporal averaging the postulated anomaly in the 1960s and 1970s in the CDK and CDM cores will be much smaller and may be due to an increased input of 210Pb in the 1970s. Such an increase has been observed already at other glaciers in the Alps, e.g. at Silvretta and Adamello (Festi et al., 2021), at Colle Gnifetti (Gäggeler et al., 1983) and at Grenzgletscher (Eichler et al., 2000) and was attributed to enhanced vertical transport related to the maximum in sulphate aerosol particles acting as transport vehicles (Eichler et al., 2000).

We regret the ambiguity in the presentation of the original version of Fig. 4. Since we were focused on the depth range corresponding to the depth-age anomaly which was sampled in relative high resolution, we only sampled the upper part for of CDK and CDM at lower frequency and interpolated the measurements. This will be changed in the revised version (showing raw data). Samples of CDK and CDM now are shown in Fig, 4 as measured and the depth intervals of the individual measurements (typically 0.7 to 1 m) are assigned clearly. In addition, as requested, we discuss whether or not the re-increase of 210Pb in the 1970s in CDK and CDM might be of atmospheric origin.

[Figure]

**Figure 4: $^{210}$Pb profiles of the three CDD ice cores. The decay-corrected $^{210}$Pb activity is shown using the drilling year of the respective ice cores as reference.** *For CDK and CDM the depths covered by the samples are plotted with thick black lines, whereas the thin integrating line is given to guide the eye and was used to calculate the $^{210}$Pb inventories.* **Blue dashed vertical lines indicate the approximate boundaries of the anomaly. Where available, absolute time markers detected over the $^{210}$Pb perturbed depth zones are also reported.** *C10 data are from Vincent et al. (1997) and this study.*

The discussion whether or not the re-increase of 210Pb in the 1970s in CDK and CDM might be of atmospheric origin was revised as follows in Section 3.2:

" *As a consequence, a strong seasonal cycle with $^{210}$Pb concentrations three to four times higher in summer than in winter is observed at high altitude Alpine sites. As expected, this also is observed in the snow deposition at CDD and shown in Fig S1 of the Supplement for summer 2004 and the outstanding hot summer 2003, for which an extremely enhanced upward transport was already reported previously (Legrand et al., 2005). Whereas in 2004 a summer*

*to winter $^{210}$Pb ratio of 2 was found, this ratio reached a factor of 7 in 2003. Together with the systematic decrease of the winter to summer layer thickness ratio with increasing core depth at the drill site (see Section 3 and Preunkert et al., 2000), this pronounced $^{210}$Pb seasonality counteracts the expected $^{210}$Pb decrease from radioactive decay.*

*Second, a well-marked anomaly characterized by $^{210}$Pb enhancements (including $^{210}$Pb peaks up to 10 times higher $^{210}$Pb than expected from atmospheric deposition) is observed in the three cores. The anomaly extends from ~83 to 108 m depth (i.e., ~26 to 54 years) in C10, ~85 to 108 m (i.e., ~32 to 70 years) in CDK, and ~82 to 102 m (i.e., ~33 to more than 58 years) in CDM ice. The $^{210}$Pb re-increase observed in CDK and CDM, however, is less pronounced than in C10. In addition, the starting depths of the CDK and CDM $^{210}$Pb re-increases correspond to the 1970s, for which $^{210}$Pb enhancements have been reported at other ice core sites (Eichler et al.,2000) and attributed to an enhanced vertical transport related to the temporal maximum of atmospheric sulfate aerosol acting as transport vehicle. To check whether these atmospheric conditions also could be responsible for the enhancement seen in CDK and CDM, we report exemplarily the CDK $^{210}$Pb activity, corrected for its respective deposition date together with the corresponding sulfate concentration in Fig. S1 of the Supplement. As mentioned above, a strong seasonality was detected in the uppermost part of the CDK core for a few years where $^{210}$Pb samples are available in seasonal resolution (Fig. S1a of the Supplement). If atmospherically derived, mean $^{210}$Pb concentrations of ice layers from 60 to 85 m depth (i.e., from 1988 to 1972), i.e., in the period for which the sulfate aerosol maximum was observed at CDD (Preunkert et al., 2001), would correspond to around 130 ± 60 mB kg$^{-1}$of $^{210}$Pb in freshly deposited snow, which is comparable to the atmospherically derived $^{210}$Pb further upward in the core. However, from 85 to 108 m depth, this connection between sulfate levels and $^{210}$Pb activity no longer holds. Whereas sulfate concentrations strongly decrease, $^{210}$Pb at the time of deposition (decay-corrected) would be strongly enhanced (mean of 600 mBq kg$^{-1}$) and far above what is expected from atmospheric $^{210}$Pb contributions. Thus, the mechanism proposed by Eichler et al. (2000) cannot be invoked in this part of the CDD core. For CDM (not shown) a similar picture appears. While from 80 to 90 m surface decay-corrected $^{210}$Pb (160 ± 70 mBq kg$^{-1}$) would not have been significantly enhanced compared to the atmospherically derived $^{210}$Pb concentrations seen further up in the CDM core, this is not the case between 90 and 103 m depth. As for CDK, mean values at the time of deposition would have been around 650 mBq kg$^{-1}$ and thus far too high to what would be expected from atmospheric transport.*

*Third, below the anomaly, a decrease ….*

Revised Fig. S1 of the Supplement now is as follows:

[Figure]

*Figure S1: 210Pb profiles of the CDK ice cores (left y-axis, black) together with corresponding SO42- concentrations (right y-axis, red). The decay-corrected 210Pb activity is shown using the from the ice core chronology estimated snow deposition date. A thin integrating line is given to guide the eye, whereas the depths covered by the samples are assigned with thick lines. The blue dashed vertical lines in (b) indicate the approximate boundaries of the anomaly.*

The entire depth records of 210Pb should be shown and not only the interval between 40 and 130 m in Fig. 4. Without the upper part, it is impossible to see if there is a decrease of 210Pb with depth at all and if the surface activity is in the range expected for glaciers in the Alps. Done. The full records will be shown in revised Fig. 4 (see above).

In the C10 core, 210Pb was determined by gamma-spectrometry (Vincent et al., 1997), whereas for the CDK and CDM cores 210Pb was analyzed by alpha-spectrometry of its decay product 210Po after chemical enrichment, which is the much more sensitive method. Gamma-spectrometry is rather insensitive due to the high conversion of the low energy gamma-line at 46 keV (96% in the form of electron and only 4% in the form of gamma-emission) and the rather low efficiency of gamma-detectors. This method is normally used for samples with high activity concentrations of 210Pb, e.g. from lake sediments. For low-activity ice samples, the

uncertainty is high (more than 50%, Vimeux et al., 2008). Especially in the region, where the anomaly was observed in the C10 core, also 137Cs activity concentrations are high due to the fallout from nuclear tests. This must have resulted in a high background in the gamma-spectrum. These uncertainties need to be discussed.

Thanks for this comment. The uncertainties of the gamma spectrometry used will be now discussed in the manuscript and respective error bars of the 210Pb data are reported. In fact, this specific gamma spectrometry method was developed at the Laboratoire de Glaciologie et Géophysique de l'Environnement, now Institut des Géosciences de l'Environnement (Delmas and Pourchet [1977], Pinglot and Pourchet, 1995) and applied in the past for many ice core studies in the Alps, sub-Arctic, South America, and also at polar sites (see e.g. Pourchet et al., 2000, Pinglot et al., 2003, Vimeux et al., 2008, Pourchet et al 2003).
Note that, since the energy of the 210Pb (46.54 keV) and the 137Cs (661 keV) is rather different (resolution of the detector is between 1.3 and 1.7 keV) we assume that the radioactive fallout from the nuclear tests should not have led to significant downgrading of the quality of the 210Pb measurements. Anyway, the adopted detection limit for C10 was assigned originally for ice core measurements which included also the time period in which 137Cs activities were high.

The following text will be added in section 2:
*"" Previously reported $^{210}$Pb measurements in C10 ice (Vincent et al. 1997) analyzed at the Laboratoire de Glaciologie et Géophysique de l'Environnement, now Institut des Géosciences de l'Environnement (IGE),were complemented by two samples measured for this study. The analytical technique was high-resolution gamma-ray spectrometry, designed to detect very low levels of radioactivity using a 20% high-purity Ge (N-type) detector, with an anti-Compton scintillation detector (Pinglot and Pourchet,1995) for which snow and ice samples were filtered previously through ion-exchange papers (Delmas and Pourchet,1977). This method is less sensitive than α-spectrometry and Vincent et al. (1997) did not assign uncertainties to their analyses. Here we estimate the uncertainty based on what has been reported in other studies using this detection method developed at IGE. Pinglot et al., 2003 reported a detection level of 10 mBq at a 97.5% confidence level for 3 days of counting on ice core samples with a typical $^{210}$Pb activity of 20 – 50 mBq kg$^{-1}$. These measurements included Chernobyl fallout in sub-Arctic glacier sites, and the levels were similar in range to the background activities of 50-100 mBq/kg found in our cores. On the other hand, detection levels of 13 and 25 mBq were calculated at 97.5 % confidence when peak interferences where neglected or considered, respectively, for a 10 g sediment sample containing 1000 times higher $^{210}$Pb activities as found in ice cores (~70 Bq kg$^{-1}$) that was measured for 63 hours (Pinglot and Pourchet, 1995). Vimeux et al. (2008) reported a lower detection limit of 4 mBq kg$^{-1}$ for $^{210}$Pb measurements (activities between 20 and 100 mBq kg$^{-1}$) on relatively small (150-250 g) ice core samples from Patagonia. The $^{210}$Pb activities in C10 ranged from 50 – 700 mBq kg$^{-1}$, with the measurements done on the C10 drilling chips merged over 3 to 5 m, allowing to obtain sample weights of up to ~ 3 to 5 kg. Since these sample masses, type (ice core sample) and geometry (filter) are comparable to those used in the Pinglot et al. (2003) study but are very different from the sediment sample in Pinglot and Pourchet (1995), we assume in the following a detection level of 10 mBq and an uncertainty of 30 mBq for the C10 $^{210}$Pb measurements. Note that, the dataset from Vincent et al. (1997) was supplemented by two additional samples for which $^{210}$Pb analysis and quality control were not yet finished in 1997.*

*….”*

It is unclear, which 210Pb decay correction was made. In Fig. 4 it is stated that for the C10 core the 1994 activity is shown. However, 210Pb activity concentration are much higher than in the original publication (Vincent et al., 1997). For a comparison between the cores, the activity should be corrected to the same reference date.

Thanks for this comment. In Fig 4, the same reference date (1994) is applied for C10 as in Vincent et al., 1997. C10 data seem to be higher in our Fig. 4 compared to the one from Vincent et al., 1997, since two additional samples were added to the data set in our study. C10 data from Vincent et al., 1997 were complemented with two additional samples for which additional measurements and quality check were done after 1997. This will be stated in the revised manuscript in section 2:

*"…Note that, the dataset from Vincent et al. (1997) was supplemented by two additional samples for which $^{210}$Pb analysis and quality control were not yet finished in 1997. ….."*

I cannot follow the argument how the presence of the crevasse caused such a large 210Pb anomaly in the C10 core, but did not affect the stratigraphy, while in the other two cores the stratigraphy was disturbed, but the 210Pb anomaly was much smaller if present at all. This is a contradiction to me.

Essentially there are two states of the crevasse – one for which the crevasse is open to bedrock and sealed by a snow bridge, and a second in which it is at least partly open to the atmosphere. Whereas in the first state the 222Rn emitted from the granite in the bedrock will accumulate, diffuse in the surrounding firn and produce 210Pb in excess (this would correspond to what is observed in C10), in the second state the excess 222Rn gas can escape from the crevasse to the atmosphere, thus 210Pb production will be strongly limited (this would correspond to what is observed in CDK and CDM). To clarify that, these two states will be reported in Fig 5b and 5c in the revised manuscript.

[Figure]

[Figure]

*Figure 5: (a) Thickness changes between 1993 and 2017. The contour lines of surface topography correspond to the 1993 surface (adapted from Vincent et al., 2020) overlain by a modelled flow line (color scale on top) which reports the calculated arrival depth at the drill site of C10, CDK, and CDM (black star) (Gilbert et al., 2014). The crevasse location (blue line) is based on the 30th June 2004 aerial photo from IGNF (see Fig.1) (b and c) Schematic representation of the origin of the 210Pb anomalies found at the drill site following the ice flow model of Gilbert et al., 2014, extracted along the flow path reaching the drill site. Isochrones are marked in red, flowlines in green (see also Section 4). The grey shaded zone indicates firn, the dotted zone indicates the snow bridge over the crevasse. Two states of the crevasse are reported: (b) the crevasse is open to the bedrock but sealed from the atmosphere by a snow bridge. In this state 222Rn and 210Pb accumulate to reach concentrations well above atmospheric conditions in the crevasse and the surrounding firn (c) the crevasse is at least partly open to the atmosphere. In this state 222Rn and 210Pb concentrations in the crevasse and the surrounding firn are strongly reduced compared to (b). The formation of missing or doubling ice layers is indicated by the orange and pink arrows.*

In addition, we will reword the discussion of this point in Section 4 as follows:

*"... "... A partial opening of the crevasse to the atmosphere would allow the bedrock-derived 222Rn in the crevasse to mix with the much lower atmospheric 222Rn concentrations (Pourchet et al., 2000). This would have led to a strong reduction of additional 222Rn accumulation and 210Pb production in the crevasse and in the snow and firn around the crevasse, starting from the moment of the opening to the atmosphere. This would explain 210Pb inventories of 70 and 55% in CDK and CDM compared to C10, because of the radioactive decay of 210Pb accumulated before the opening of the crevasse to the atmosphere, over 10 and 18 years, respectively....."*

The agreement between the nitrate records obtained at a nearly identical location (please add coordinates to support this statement) is not as good as I would expect. Maybe plotting them against a m water equivalent scale would make it easier to identify common features. Generally, I find Fig. 2 difficult and confusing. What are the 250 ppb and 400 ppb levels?

The GPS coordinates of the ice cores were checked. Differences are at most 10m distance. We will provide the mean GPS coordinates of the three cores in the Introduction:

*"…Underpinning these efforts are three ice cores all drilled to bedrock within maximal 10 m of each other (mean geographic location of 45.842195° N, 6.84675° E) in 1994 …"*

The fact that the agreement of the NO3 depth profile is not as good as expected by the reviewer is likely due to the fact that the corresponding layers were not deposited at the same location along the flowline since the cores were not drilled at the same time. E.g., the layer of 1990 is in C10 at 25 m depth and in CDM at 67 m depth. As shown in Fig. 5a, the deposition locations were different by around 50 m. Taking in account the changing accumulation and winter to summer deposition ratio upstream the core (see section 3) this would result in stratigraphic differences in the NO3 (and all ion) depth profiles.

To point this out we added in the caption of Fig. 2 the following sentence:

*"…Note that the chronological changes of the NO3 concentrations are offset in depth relative to each other due to the different years the cores were drilled."*

We prefer to keep m scale in Fig. 2 since with that the reader can directly compare the depths in the core with the modelled surface deposition sites of the ice layers (see Fig. 5a). In any case, since ice core sections reported in Fig. 2 (except for the upper part of C10) are below the close off, which is around 50 m depth at the drill site, overplotting the m water equivalent scale would not change the picture.

The bars of 250 and 400 ppb were removed in Fig. 2 since they are not used in the discussion.

What is also puzzling is that 14% of the nitrate values (and even 30% of the ammonium data) were discarded. What is the basis for that? Which criteria did you use to identify contaminated values?

Thanks for this remark. After careful consideration we found that we made an error in the original manuscript. In fact much less data were discarded due to contamination. This is explained now in more detail.

In section 2 it will read:

*"….Despite the undersized core section available for the CFA analyses at CEP, 86% of the ice core could be analyzed. The nitrate profile obtained at DRI and CEP (covering 97% in this depth range), were compared from 45 to 86 m depth. Both datasets are in very good agreement (Fig. 2). After having additionally discarded very high peaks in $NO_3^-$ values (1.5% of CEP data), which were not present in the DRI dataset and could be attributed easily to contamination, mean $NO_3^-$ values from 45.3-86.0 m were 263 ppb (CEP) and 255 ppb (DRI). The agreement is somewhat weaker for $NH_4^+$ likely because only 80% of the depth range is covered by the CEP measurements. After discarding additionally 8 % of the CEP $NH_4^+$ data consisting of high $NH_4^+$ peaks which were not present in the DRI dataset, the mean $NH_4^+$ values of 101 ppb (CEP) and 95 ppb (DRI) were in good agreement. ....."*

Why are the tritium records not continuous? With a discontinuous record it is difficult to identify the 1963 maximum. In the case of the CDK core the maximum might be at 86 m.

The CDK and CDM records were mainly dated first using the depth stratigraphy of major ions and by comparison with C10, and only the depth range over which the 3H bomb test maximum was expected was analyzed for tritium. This is common approach when searching the 1963 bomb maximum since the rest of the 3H depth profile is rather uninteresting scientifically.

In the case of CDK, ice layers at 86m depth could be clearly assigned to the years 1970 (see Legrand et al., 2013 and Figs. 2 and 3). Thus, in our opinion there is no reason to search for the 1963 bomb horizon at this core depth.

**Minor comments**

Bachelor thesis's cited (Waldner, Zipf) are not publicly available. Include information in supplement.

The references for the two Bachelor thesis were initially put to credit the work of the two students, but finally both became co-authors. The references were deleted, and their work (analyses of the samples) will be credited in the author contribution section. Analytical details on measurements are already published and referenced in the manuscript (see Section 2). The detection limit appropriate to the method used in the CDK and CDM 210Pb analyses will be given Section 2 in the revised manuscript:

"$^{210}$Pb samples of CDK and CDM ice were analyzed at IUP by α-spectrometry for its decay product $^{210}$Po. Typical blank values of (5.7 ± 2.5) $10^{-5}$ Bq for $^{210}$Po and (3.8 ± 1.6) $10^{-5}$ Bq for $^{209}$Po were subtracted from the sample counts (see Stanzick, 2001, and Elsässer et al., 2011 for further working analytical conditions)."

Thus, we feel that no supplement information is necessary.

103: Despite the undersized core section available at CEP, the nitrate profile obtained at DRI and IUP are in very good agreement (Fig. 2). Do you mean DRI and CEP?

Thanks, CEP was meant, this was corrected.

177: How was the winter to summer layer thickness ratio obtained?

The winter to summer layer thickness ratio was calculated on the basis of the NH4 depth stratigraphy. Details can be found in Preunkert et al., 2000. This now is assigned clearly in the revised manuscript when the term "winter to summer layer thickness" first appears (Section 3.1).

"…and the winter to summer layer thickness ratio, calculated on the basis of the ammonium depth stratigraphy (see details in Preunkert et al., 2000), decreases from 1 at the surface to 0.5 at 100m depth. "

216:  Result of annual layer counting, what do you mean with that?

Thanks for this remark, we clarified this sentence in section 3.1.

*"…. As a consequence, annual layer thicknesses of only 0.7 and 0.2 mwe are observed at 100 m and 118 m depth (Preunkert et al., 2000) and the winter to summer layer thickness ratio, calculated on the basis of the ammonium depth stratigraphy (see details in Preunkert et al., 2000), decreases from 1 at the surface to 0.5 at 100 m depth….."*

239: For 210Pb seasonality include reference Eichler et al., 2000.
ok done

*"…CDM $^{210}$Pb re-increases correspond to the 1970s, for which $^{210}$Pb enhancements have been reported at other ice core sites (Eichler et al.,2000) and attributed to an enhanced vertical transport related to the temporal maximum of atmospheric sulfate aerosol acting as transport vehicle..;"*

245-250: A zero 210Pb level can only be seen if the values are blank corrected and if the ice does not contain any supported 210Pb from mineral dust (see e.g. Gäggeler et al., 2020). Did you do a blank correction and what was the blank?
Thanks for your remark, the data are blank corrected. The "non-zero" term will be changed in "above detection limit" and the detection limit will be added in the manuscript in Section 2.
*" $^{210}$Pb samples of CDK and CDM ice were analyzed at IUP by α-spectrometry for its decay product $^{210}$Po. Typical blank values of (5.7 ± 2.5) $10^{-5}$ Bq for $^{210}$Po and (3.8 ± 1.6) $10^{-5}$ Bq for $^{209}$Po were subtracted from the sample counts (see Stanzick, 2001, and Elsässer et al., 2011 for further working analytical conditions)…."*

and in Section 3.2 it will read:
*"..However, it is worth noting that, especially in the case of the CDM and CDK cores, $^{210}$Pb activity (after blank correction) is above detection limits even in the bottommost core sections, while in C10 levels are below the detection limit…"*

Figure 3: C10 was drilled in 1994. Why do the records of annual layer thickness and nitrate concentration continue to the year 2000?
Thanks for this remark. We agree that the Fig. and caption needed improvement. C10 and CDM annual layer thickness data are shifted in time to compensate for the different drilling dates of the three ice cores. The revised Fig. 3 is as follows:

[Figure]

**Figure 3: (a) Annual layer thickness of C10 (Preunkert et al. 2000) and CDK (Legrand et al., 2013) compared to CDM. To compensate for the different drilling dates of the three cores, annual layer thickness data of C10 and CDM were shifted for +10 and -8 years, respectively. For CDM, the annual layer thickness is estimated via the ammonium stratigraphy back to 1980 and via the nitrate (and ammonium) stratigraphy further back in time (Section 3.1.3). (b) comparison of nitrate summer half-year means of C10 (Preunkert et al., 2003), and CDK (Legrand et al., 2013) with CDM. The thick solid lines for C10 and CDK refer to the smoothed profile (single spectrum analysis, see Legrand et al., 2013). CDM depth intervals for which the dating is uncertain (Section 3.1.3), are marked with dashed lines.**

**References**

Eichler, A., Schwikowski, M., Gäggeler, H.W., Furrer, V., Synal, H.-A., Beer, J., Saurer, M., Funk, M., 2000. Glaciochemical dating of an ice core from upper Grenzgletscher (4200 m a.s.l.). Journal of Glaciology 46, 507-515.

Festi, D., Schwikowski, M., Maggi, V., Oeggl, K., Jenk, T.M., 2021. Significant mass loss in the accumulation area of the Adamello glacier indicated by the chronology of a 46 m ice core. The Cryosphere 15, 4135-4143.

Gäggeler, H.W., Tobler, L., Schwikowski, M., Jenk, T.M., 2020. Application of the radionuclide 210Pb in glaciology – an overview. Journal of Glaciology 66, 447-456.

Vimeux, F., de Angelis, M., Ginot, P., Magand, O., Casassa, G., Pouyaud, B., Falourd, S., Johnsen, S., 2008. A promising location in Patagonia for paleoclimate and paleoenvironmental reconstructions revealed by a shallow firn core from Monte San Valentín (Northern Patagonia

Icefield, Chile). Journal of Geophysical Research: Atmospheres 113, D16118.

---

## Referee Report (RR1)

Alison Criscitiello
Review of: "Impact of subsurface crevassing on the depth-age relationship of high- alpine ice cores extracted at Col du Dôme between 1994 and 2012"

I will keep this third review brief in the interest of expediency. Preunkert et al. compare records from three ice cores from Col du Dôme drilled in 1994, 2004, and 2012. The 1994 (C10) age scale is intact, while the 2004 (CDK) and 2012 (CDM) age scales are disturbed during the 1950s/60s. Chronologies are largely established by annual layer counting of ammonia, with the disturbances identified by the $H_3$ and $^{137}Cs$ records. Disturbances are attributed to the presence of an upstream (to flow) crevasse, which did not intersect the flow path reaching the core site in 1994 (but it did subsequently). During times when a snow bridge covers the top of the crevasse, $^{210}Pb$ accumulates in the crevasse and surrounding firn. I find the discussion and suggested mechanisms for both the observed layer doubling (or missing layers) and the impact of the two crevasse states on $^{222}Rn$ and $^{210}Pb$ accumulation novel and fairly well supported. This is an interesting and new theory for processes occurring at this site, and I enjoyed reading the mss.

In the revised mss, I can see huge improvements in the edits and additions made in response to the previous referee and editor reviews. Really like the added Fig.2 b and c panels. A few questions and comments below.

- I think there was a core drilled on Dôme de Gouter, correct? Is there an archive stick left, that perhaps $^{210}Pb$ could be measured on (for comparison to CDD)?
- Fig. 2 is rather confusing. Not sure how to simplify it or make it more readable. Perhaps the insets could be moved to a separate figure? Worth a think. I don't like the $NO_3$ offset which is an artifact of drilling year, I understand, but could perhaps be corrected for so the records line up?
- "In the C10 core, $^{210}Pb$ was determined by gamma-spectrometry (Vincent et al., 1997), whereas for the CDK and CDM cores $^{210}Pb$ was analyzed by alpha-spectrometry of its decay product $^{210}Po$ after chemical enrichment, which is the much more sensitive method." This is one of the more concerning aspects of the mss I found. Uncertainties arising from utilizing $^{210}Pb$ data obtained by more than one method should be discussed in more detail.
- "After having additionally discarded very high peaks in $NO_3^-$ values (1.5% of CEP data), which were not present in the DRI dataset and could be attributed easily to contamination, mean $NO_3^-$ values from 45.3-86.0 m were 263 ppb (CEP) and 255 ppb (DRI). The agreement is somewhat weaker for $NH_4^+$ likely because only 80% of the depth range is covered by the CEP measurements. After discarding additionally 8 % of the CEP $NH_4^+$ data consisting of high $NH_4^+$ peaks which were not present in the DRI dataset, the mean $NH_4^+$ values of 101 ppb (CEP) and 95 ppb (DRI) were in good agreement." I find the discarding of what amounts to quite a lot of data concerning. Was a threshold technique used? How did you determine that clear contamination had occurred? You assume the DRI CFA is the benchmark, and any large deviations that don't align with that record must be contamination? It currently reads as a bit subjective.

- **Was** the winter to summer layer thickness ratio obtained just from ammonium? Were other glaciochemistry time series used as well?
- "The dating of the C10 ice core back to 1925 obtained from annual layer counting of the ammonium record was initially established by Preunkert et al. (2000). More recently, the availability of additional measurements such as lead, cadmium and thallium allowed the dating to be extended back to 1890 without changing the original dating back to 1935 (Legrand et al., 2018)." This implies (with no mention of it) that you changed the original dating between 1925 and 1935, yes? Maybe say a bit more about this (how did you identify a dating error? etc).
- The bomb test horizon insets shown in Fig.4 are really lacking. I certainly understand only looking in certain sections of the core for bomb horizons (where they're expected), but why are there so few measurements? There are so few that it isn't actually possible to confidently pick 1963 (or <1954) at CDM or CDK. Were only wings (bag averages) available?

---

## Author Response (AR2)

Reponses to reviewers' comments.

Please note, that Figure numbers and Sections given in the following refer to the manuscript as it was revised in May 2023. If they would change in the potential next manuscript version this is assigned "now Fig. X" and or "now Section X". Line numbers refer to the new version of the manuscript.

**Reviewer 1**
Thanks for having reviewed our manuscript a second time.

The caption of Figure 1 could use more information - can the arrows be color coded such that portion of the crevasse seen in both a) and b) are the same, and the lighter arrows then define extension of the crevasse in a)? I find it hard to tell what portion of the feature is the same in the two photos, and what portion of the feature is new in a) compared to b)

Thanks for your comment. We agree that it is hard to bring both photos together since they were not taken from the same position. a) was taken from the helicopter and b) from the slope which leads to the Vallot Observatory. We color coded common crevasse features as you advised (see figure and caption below).

- describe the difference between the solid and dashed line in c) in the caption
thanks, this was done (see figure caption below)

- c) looks like a topographic map to me the caption describes it as an aerial photograph. i think it should be reworded to indicate that the topographic map is from aerial photographs
Yes, c) is a topographic map of the glacier surface and sorry if this was not clearly identified in the caption. We've also the bedrock topography in Figure 1c (now Figure 2c). The caption is updated now and clearly states that c) is a topographic map. This part of the caption now reads:
*(c) Topographic map of the Col du Dome and Dome de Gouter together with the underlying bedrock topography (adapted from Vincent et al., 2020). Contour lines are spaced at 5 m intervals.*

[Figure]

**Figure 2: View of the South-East flank of the Dome de Gouter and Col du Dome saddle including the drill site of 1994, 2004, and 2012 situated downslope of Dome du Gouter. Note that the three drill sites are located within about 10 m of each other and thus are indicated by a single dot in (c). (a) Picture taken in summer 2012: A large crevasse extends across the upstream catchment area of the drilling site. At the time of the picture the distinctly visible crevasse was mainly snow-covered. A potential second crevasse also is visible on the southwestern slope of the glacier. (b) Picture taken in summer 1999: Evidence of one to two crevasses limited to the southwestern side of the Dome du Gouter. Black arrows in a) and b) indicate parts of the crevasse which are suggested by the surface features in a) and b). Grey arrows in a) mark the part of the crevasse which was only visible in 2012 (c) Topographic map of the Col du Dome and Dome de Gouter together with the underlying bedrock topography (adapted from Vincent et al., 2020). Contour lines are spaced at 5 m intervals. The main crevasse highlighted in (a) and (b) is reported in (c) based on an aerial photo from Institut national de l'information géographique et forestière (IGNF) taken at 30th June 2004 (blue solid line in (c) indicates the part of the crevasse which was clearly visible, and blue dashed lines demark the part which was less clearly visible).**

Thanks for having reviewed our manuscript a second time.

The responses to my comments are not satisfying. There are again a number of scientific inconsistencies and lack of scientific thoroughness as outlined below, which prevent assessing the scientific quality.

210Pb lead record: The spatial resolution for most of the samples was included now, but this is not sufficient. A consistent temporal averaging has to be done to ensure data comparability. Now additional questions arose: 1) why is the spatial resolution not shown for all samples? 2) Why are there two data points at 60 m depth of CDM?

Thanks for this comment. We agree in principle that consistent temporal averaging would be good to compare the $^{210}$Pb records. However, for our application and in view of the small depth coverage in the upper part of CDK and CDM this is of no advantage. We clarified in the text that $^{210}$Pb measurements were focused on the part where the $^{210}$Pb profile is disturbed and that in the upper part only sporadic $^{210}$Pb measurements are available.

In Section 2 (now Section 2.2 line 199) it reads:
*… We stress that whereas $^{210}$Pb was measured continuously on discrete samples covering the whole C10 ice core, $^{210}$Pb measurements in CDK and CDM were focused on the $^{210}$Pb anomaly starting around 80 m depth. Therefore, only point-wise measurements with sample lengths between 0.6 to 1m length, i.e. covering less than one year, were made in the upper part of the latter two cores, with the exception of two CDM samples which were integrated of over core depths of 10 m each (covering 2 and 4 years).*

And in of Section 3.2, line 393:
*We stress that whereas $^{210}$Pb was measured continuously on the C10 ice core, $^{210}$Pb measurements on CDK and CDM were focused on the $^{210}$Pb anomalies starting below 80 m. Therefore, only a few samples with limited depth and time coverage are available in the upper parts of CDK and CDM. However, comparing $^{210}$Pb levels of the shorter CDM samples with the two samples integrating 2 and 4 years, and in view of the limited seasonal variation (with the exception of the outstanding hot summer of 2003) observed in Fig. 7, we assume the sample lengths and coverage is good enough to depict the order of magnitude of $^{210}$Pb activities in the upper parts of the cores. …..*

Concerning the two questions of this comment:
1) As outlined in the Fig. 5 caption (now Fig. 6) (*…For CDM (a) and CDK (b) the depths covered by the samples are plotted with thick black lines, whereas the thin lines are given to guide the eye and were used to calculate the $^{210}$Pb inventories…..*)
the spatial extension of all CDM and CDK samples was added with thick lines. Since typically ice core sections of 0.6 to 1.0 m long ice core sections were analyzed the spatial extension is not very large for some samples.

2) For CDM, in addition to the samples with a length ranging between 0.6 and 1.0m, two parallel samples were measured integrating several meters of depth. This was done to get an idea of the representativeness of the shorter samples.
This is explained now in the manuscript, together with the fact that the $^{210}$Pb measurements in CDK and CDM were focused only on the parts of the cores where the $^{210}$Pb profiles were disturbed.

In Section 2 (now Section 2.2 line 199) it reads (as above):
*… We stress that whereas $^{210}$Pb was measured continuously on discrete samples covering the whole C10 ice core, $^{210}$Pb measurements in CDK and CDM were focused on the $^{210}$Pb anomaly starting around 80 m depth. Therefore, only point-wise measurements with sample lengths between 0.6 to 1m length, i.e. covering less than one year, were made in the upper part of the latter two cores, with the exception of two CDM samples which were integrated of over core depths of 10 m each (covering 2 and 4 years)…*

And in Section 3.2. line 393 it reads (as above):
*We stress that whereas $^{210}$Pb was measured continuously on the C10 ice core, $^{210}$Pb measurements on CDK and CDM were focused on the $^{210}$Pb anomalies starting below 80 m. Therefore, only a few samples with limited depth and time coverage are available in the upper parts of CDK and CDM. However, comparing $^{210}$Pb levels of the shorter CDM samples with the two samples integrating 2 and 4 years, and in view of the limited seasonal variation (with the exception of the outstanding hot summer of 2003) observed in Fig. 7, we assume the sample lengths and coverage is good enough to depict the order of magnitude of $^{210}$Pb activities in the upper parts of the cores. …..*

The new statement in the revised text that together with the systematic decrease of the winter to summer layer thickness ratio with increasing depth, seasonality counteracts the expected 210Pb decrease from radioactive decay is critical. If this decrease in winter snow contribution resulted in increased 210Pb values, how can this be disentangled from the evoked crevasse effect?

The winter to summer snow ratio versus depth at the drill site is detailed in Preunkert et al., 2000, Figure 9. The ratio decreases from ~1 at the surface to 0.6 at 75 m depth (well above the $^{210}$Pb disturbance). This decrease of winter to summer ratio by almost a factor of 2 counteracts the decrease of $^{210}$Pb, as stated in the text in Section 3.2. Below 75m and down to 111 m (i.e., over the depth of the $^{210}$Pb anomaly), the winter to summer ratio does not change significantly but stays between 0.6 and 0.5.
Therefore, it will not contribute significantly to the $^{210}$Pb enhancement observed in the cores below 80m.

This also is detailed in the revised text in Section 3.2 line 433:
*… Together with the systematic decrease of the winter to summer layer thickness ratio with increasing core depth down to 75 m (from 1 to 0.6) at the drill site (see Sect. 3 and Preunkert et al., 2000), this pronounced $^{210}$Pb seasonality at least partly counteracts the expected $^{210}$Pb decrease from radioactive decay. ….*

*and line 449:*

*... Since winter to summer snow ratios lie consistently between 0.5 and 0.6 in the C10 core in the depth interval of the $^{210}$Pb anomaly (Preunkert et al., 2000), these increases in $^{210}$Pb cannot be attributed to changes in seasonal snow deposition.*

The discussion on detection limits and uncertainties of 210Pb analyses with gamma spectrometry is still confusing. Both should be given in the same unit as the 210Pb values shown in Fig. 4 (mBq kg-1). In the new text both are given in mBq. The same is true for the blank values of the alpha-spectrometry. This is given in Bq. Without any information on samples size and counting time, this cannot be related to Fig. 4.

Thanks for this comment. We think you meant Fig. 5 now Fig. 6.
The blank values of $^{210}$Pb and 3H measurements are given now in Bq kg-1 and TU, respectively.

The text in Section 2 (now Section 2.2) concerning the $^{210}$Pb now reads as follows for the measurements made at IUP (line 165):
*.... With typical sample masses of 300 to 1,000 g and measurement times of 2 to 6 days, mean $^{210}$Pb measurement errors of 4 ± 4 mBq kg$^{-1}$ were achieved on ice core drill chip samples spanning ice core depths between 0.6 and 1 m.*

And for the $^{210}$Pb data from C10 measured at IGE (line 192):
*... we assume in the following a detection level of 10 mBq assigned by Pinglot et al. (2003) and a maximum uncertainty of 30 mBq for all C10 $^{210}$Pb measurements. Taking 1 kg sample mass as an absolute lower limit, this would amount to a total error of 30 mBq kg$^{-1}$.*

In addition, it is now stated that two additional 210Pb values of C10 were added to Fig. 4, which were not published in Vincent et al., 1997. Those are just the two highest values in the entire record. It would be more convincing to show the comparison between the inner and outer samples. The same is true for the separation of 137Cs and 210Pb: it would be convincing to see a spectrum. Especially because the C10 core is the one for which the 210Pb artefact is most obvious and that was analysed with the less sensitive method.

The reason why these high values were added after the publication of Vincent et al. (1997) is already given in the text in Section 2 (now Section 2.2). The method to measure $^{210}$Pb with gamma detection was well established in the literature 30 years ago and is therefore referenceable. It is clearly beyond the scope of this manuscript to provide the analytical details here. In addition, $^{210}$Pb values are not extremely high if compared to the $^{210}$Pb peak values detected with alpha spectrometry in the CDK and CDM cores (measured by alpha spectrometry). Pinglot and Pourchet (1995) compared alpha and gamma spectrometry measurements of a sediment sample and achieved a fairly good agreement of both measurements. This comparison is now reported in the manuscript, and when comparing the $^{210}$Pb inventories in Section 3.2 the difference in the two methods is noted.

This reads in the text as follows in Section 2 (now Section 2.2 line 174):

*Although made on a sediment sample with much higher specific $^{210}$Pb activities than found in core cores, Pinglot and Pourchet (1995) made a direct comparison of $^{210}$Pb alpha and gamma-ray measurements. They found that the measurements made with α-spectrometry were only ~84 ± 11 % of the respective values obtained with gamma spectrometry and attributed the difference to insufficient acid leaching during the α-spectrometry sample preparation. However, both methods generally provide comparable activity values and the relative temporal variations in the activities should be robust. …*

and Section 3.2. line 493:
*… Furthermore, the CDK and CDM $^{210}$Pb anomaly inventories (Fig. 6) are 4 times lower than in C10, which cannot be explained by analytical differences of the measurement methods (see Sect. 2.2). We stress that the ice originating at the crevasse had essentially the same travel time from the crevasse to the drill site for a given depth in case of all three cores….*

The hypothesis of the two states of the crevasse is not supported by the data. The upper parts of the nitrate records (1990-1979) show the expected compression/thinning from 1994 to 2012 with a decrease in depth range covered by those 11 years from 35 m (1994), 21 m (2004) to 14 m (2012). However, the depth range from 71 m to 79 m, i.e. 8 m in C10 (1994) corresponds to the depth range of 78 m to 89 m, i.e. 11 m in CDK (2004) and probably 88 m to 96 m, i.e. 8 m in CDM (2012). Such a behaviour is highly unlikely for cores collected only 10 m apart from each other, since it would imply not only no thinning, but on the contrary thickening of CDK. This leaves doubts about the dating and also about the decay-corrected 210Pb activity in Fig. S1, which depends on an exact dating.

Thanks for this comment. We added a paragraph in Section 3.1.2 about that inhomogeneity between CDK and C10. In fact, a similar feature also was observed in CDM (Section 3.1.3). As in CDM this annual layer thickness increase is just at the starting limit of the discontinuity in CDK, which might have started already a few meters above 89 m depth, i.e. at 85-87m, just at the depth where the $^{210}$Pb rises.
The resulting dating uncertainty (see text in Section 3.1.2) has, however, a very limited effect on Fig. S1 (now Fig. 7) since there is no $^{210}$Pb datapoint in the range of 85 to 87m, i.e. the depth interval for which dating is more uncertain in the case that these snow layers were not integrated via regular surface deposition in the ice core.

The text in Section 3.1.2 (line 301) was changed accordingly:
*A closer look at the $NO_3^-$ and $NH_4^+$ raw data in CDK shows that the depth interval from 80 to 89 m appears to correspond to 72 to 79 m depth in C10 and may be attributed to the years 1976-1971 as done by Legrand et al. (2013). However, these 5 years span 2 m more in CDK than in C10 corresponding to a relative thickening of layers by a factor of 1.28. Such an anomaly in the thinning curve cannot be explained by the systematic layer thinning at the drill site caused by undisturbed upstream inflow of ice (see also Fig. 3) along the same flow line for CDK and C10. However, refilling the void of the crevasse by inflow of ice from upstream may explain such a thickening.*

and to Figure caption of Figure S1 (now Fig. 7) we added:

*… To avoid a potential overestimation of $^{210}$Pb activities between 85 and 92 m due to the dating uncertainty, the recent date limits of the uncertainties were assigned to the samples..*

Discarding nitrate and ammonium data: The procedure of removing CEP peaks because they were not present in the DRI dataset is not convincing and not scientifically sound. Why is the DRI assumed to be correct? There are still CEP values in the figure, which are higher than the corresponding DRI value. You need an independent criteria to identify contaminated values. In addition, most of the data gaps are in the sections below 88 m, for which no DRI data are shown. What is the reason for those data gaps?

Thanks for this comment.
First, the reason for the data gaps below 88 m is that the ice core sections available at CEP were too small for reliable analyses.

We tried to better explain this and the reasoning underpinning the data comparison between DRI and CEP.

DRI dataset is assumed to be correct since working conditions were as they should be, i.e. the ice section had the regular size for routine CFA measurements using the DRI analytical system. This was not the case at CEP since the available ice core section for CFA measurements were at or below the lower size limit for the analytical system and, given the small cross section, we were not completely confident that the sample had not been altered by circulating laboratory air.

Taking the advantage of having a second dataset measured under regular conditions over a limited depth interval, we made a comparison of the two datasets in the depth interval covered by both of them, to get an idea whether the analyzed CEP data are reliable and useful for their application in this study. Note that further down in the CDM ice core where no DRI data were available, no additional datapoints were discarded from the CEP dataset. However, we used the CEP DRI comparison to conclude on the reliability of NO3 and NH4 data measured in these core depths. We state this now clearly in the text.

The corresponding text in Section 2 (now Section 2.2 (line 215)) reads now as follows:
*… However, since the CDM ice core has only a 3-inch diameter, the ice available for the CFA analyses at CEP consisted only of a non-rectangular cross-section with maximum outer dimensions of 2.5 x 3.0 cm instead of the standard quadratic size of 3.5 x 3.5 cm for which the standard melt head at CEP is designed. Although a special, smaller melt head was constructed for the CDM analyses, it was not always possible to assure that the CFA melt water only came from an inner section of the ice material with no contact to the outer surfaces. This may have led to a higher risk of contamination of the inner sample melt water stream and with the smaller melt water flow available implied also a reduced analyte spectrum. Despite the undersized core section available for the CFA analyses at CEP, 86% of $NO_3^-$ and/or $NH_4^+$ raw data could be evaluated. To test the reliability of the CEP dataset, the nitrate profiles obtained at DRI and CEP (covering 97% in this depth range) were compared over the depth interval 45 to 86 m. Both datasets are in very good overall agreement, except for individual outliers in the CEP data.*

*After having additionally discarded very high peaks (concentrations above 700 ppb) in $NO_3^-$ values (1.5% of CEP data in the depth interval from 45 to 86 m), which were not present in the DRI dataset and could be attributed easily to contamination, mean $NO_3^-$ values over this depth interval were 263±281 ppb (CEP) and 255±231 ppb (DRI) (Fig. 4). The agreement is somewhat weaker for $NH_4^+$ likely because for this species only 80% of this depth range is covered by CEP measurements. After discarding additionally 8 % of the CEP $NH_4^+$ data between 45 and 86 m consisting of high $NH_4^+$ peaks (concentrations exceeding 190 ppb), which were not present in the DRI dataset, the mean $NH_4^+$ values of 101±110 ppb (CEP) and 95±99 ppb (DRI) were in good agreement. As a consequence of the better reliability, we base our discussion mainly on the $NO_3^-$ data. Below 86 m no additional data were discarded from the CEP $NO_3^-$ and $NH_4^+$ datasets. However, because no further single $NO_3^-$ peak values above 700 ppb were found below 86 m, we are confident in $NO_3^-$ data below this depth. In the case of $NH_4^+$ we cannot exclude that a few peaks in the record below 86 m with a concentration higher than 200 ppb might be influenced by contamination.*

Figure 3: I still think that you cannot show any data for C10 beyond the year of drilling (1994), because they don't exist. Shifting does not help.
We think you meant Fig. 4 (now Fig. 5). It makes no sense to compare annual layer thicknesses of the three ice cores on absolute age if they are not drilled in the same year. Thus, to overcome this problem we shifted the data of CDM and C10 in time so that all three data sets to simulate a common drilling year. This also was stated in the legend. Since this was not clear to the reviewer, we changed the vertical scale in Fig 4a to "Year before drilling".

Note that to increase data consistency in recent and the present publications we updated Figure 4 (without changing its scientific meaning) to no longer show the originally published data versions of Legrand et al., 2013 and Preunkert et al., 2003, but the actual data as they are archived in the database.

[Figure]

**Figure 5: (a) Annual layer thickness of C10 (adapted from Preunkert et al. 2000), CDK (adapted from Legrand et al., 2013) and CDM. For CDM, the annual layer thickness is estimated via the ammonium stratigraphy back to 1980 and via the nitrate (and ammonium) stratigraphy further back in time (Sect. 3.1.3). (b) comparison of nitrate summer half-year means of C10 (adapted from Preunkert et al., 2003), CDK (adapted from Legrand et al., 2013) and CDM. The thick solid lines for C10 and CDK refer to the smoothed profiles (single spectrum analysis, see Legrand et al., 2013). CDM depth intervals for which the dating is uncertain (Sect. 3.1.3), are marked with dashed lines.**

Confusing is that in the revised version new data from another core from 1994 are shown additionally in Fig. 5, which are not mentioned in Tab. 1 and which have a different labeling than in Vincent et al., 1997.

Thanks for this comment, sorry that we overlooked that.
Data in Table 1 are limited to the three cores which could be investigated in detail.

We updated the title of Table 1 accordingly to:
*Table 1: Basic glaciological and radiometric parameters of the three CDD ice cores investigated in this study.*

The Fig. 5 caption (now Fig. 6) was updated to read as follows:
*… are compared to those from a 140 m long ice core extracted 30 m away from C10 also in 1994 (labeled as "ice core 2" in Vincent et al., 1997 and denoted here as C11, upper x-axis in c, orange)…*

Alison Criscitiello
Review of: "Impact of subsurface crevassing on the depth-age relationship of high-alpine ice cores extracted at Col du Dôme between 1994 and 2012"
I will keep this third review brief in the interest of expediency. Preunkert et al. compare records from three ice cores from Col du Dôme drilled in 1994, 2004, and 2012. The 1994 (C10) age scale is intact, while the 2004 (CDK) and 2012 (CDM) age scales are disturbed during the 1950s/60s. Chronologies are largely established by annual layer counting of ammonia, with the disturbances identified by the H3 and 137Cs records. Disturbances are attributed to the presence of an upstream (to flow) crevasse, which did not intersect the flow path reaching the core site in 1994 (but it did subsequently). During times when a snow bridge covers the top of the crevasse, 210Pb accumulates in the crevasse and surrounding firn. I find the discussion and suggested mechanisms for both the observed layer doubling (or missing layers) and the impact of the two crevasse states on 222Rn and 210Pb accumulation novel and fairly well supported. This is an interesting and new theory for processes occurring at this site, and I enjoyed reading the mss.

In the revised mss, I can see huge improvements in the edits and additions made in response to the previous referee and editor reviews. Really like the added Fig.2 b and c panels. A few questions and comments below.

- I think there was a core drilled on Dôme de Gouter, correct? Is there an archive stick left, that perhaps 210Pb could be measured on (for comparison to CDD)?
  Thanks for this comment. In principle this would be a good idea. However, there is not much material left and the advantage we could get from that will be very limited since there are additional crevasses at the Dome de Gouter itself and very high $^{210}$Pb values (28 Bq k-1) are reported by Vincent et al., 1997.
  We give this value now also in the text and have added a paragraph in which we roughly estimate whether the crevasse could be responsible for the enhanced $^{210}$Pb values observe in the ice cores investigated here.

- Fig. 2 is rather confusing. Not sure how to simplify it or make it more readable. Perhaps the insets could be moved to a separate figure? Worth a think. I don't like the NO3 offset which is an artifact of drilling year, I understand, but could perhaps be corrected for so the records line up?
  We think you meant Fig. 3 (now Fig. 4). We are not totally sure what is meant with NO3 offset, there is an offset in the depth scale we have to apply because of the different drilling years. We tried to rearrange the Figure so that there are common depth scales for all three ice cores. However, this would crush the depth zones around the $^{210}$Pb disturbances which are of main interest in this manuscript. We puzzled also about the 3H/137Cs insets in the figure, but in the end we found its best to leave them as is. Among others also since in the discussions e.g. Section 3.1.2 and 3.1.3 the 3H und 137Cs markers are used directly in connection with the ionic stratigraphies. Please note also that the annual layer thicknesses of the three cores, for which a comparison in Fig.3 (now Fig.4) is not obvious are reported in Fig. 4 (now Fig. 5).

- "In the C10 core, 210Pb was determined by gamma-spectrometry (Vincent et al., 1997), whereas for the CDK and CDM cores 210Pb was analyzed by alpha-spectrometry of its decay product 210Po after chemical enrichment, which is the much more sensitive method." This is one of the more concerning aspects of the mss I found. Uncertainties arising from utilizing 210Pb data obtained by more than one method should be discussed in more detail.

  Thanks for this comment. This matter is now discussed in more detailed including a comparison between the gamma ray method used for the C10 measurements and a parallel measurement with alpha spectrometry. The differences amount to about 20%. We point out this difference when comparing the $^{210}$Pb inventories of the three ice cores.

  The text in Section 2 (now Section 2.2 line 172) reads:
  *… We note that the gamma-ray method is less sensitive than α-spectrometry due to the high conversion of the low energy γ-line at 46 keV (96% in the form of electron and only 4% in the form of γ emission) (Gaeggeler et al.2022), and may have systematic differences. Although made on a sediment sample with much higher specific $^{210}$Pb activities than found in core cores, Pinglot and Pourchet (1995) made a direct comparison of $^{210}$Pb alpha and gamma-ray measurements. They found that the measurements made with α-spectrometry were only ~84 ± 11 % of the respective values obtained with gamma spectrometry and attributed the difference to insufficient acid leaching during the α-spectrometry sample preparation. However, both methods generally provide comparable activity values and the relative temporal variations in the activities should be robust.…*

  The text in Section 3.2 (line 494) reads:
  *… Furthermore, the CDK and CDM $^{210}$Pb anomaly inventories (Fig. 6) are 4 times lower than in C10, which cannot be explained by analytical differences of the measurement methods (see Sect. 2.2). We stress that the ice originating at the crevasse had essentially the same travel time from the crevasse to the drill site for a given depth in case of all three cores. .…*

- "After having additionally discarded very high peaks in NO3- values (1.5% of CEP data), which were not present in the DRI dataset and could be attributed easily to contamination, mean NO3- values from 45.3-86.0 m were 263 ppb (CEP) and 255 ppb (DRI). The agreement is somewhat weaker for NH4+ likely because only 80% of the depth range is covered by the CEP measurements. After discarding additionally 8 % of the CEP NH4+ data consisting of high NH4+ peaks which were not present in the DRI dataset, the mean NH4+ values of 101 ppb (CEP) and 95 ppb (DRI) were in good agreement." I find the discarding of what amounts to quite a lot of data concerning. Was a threshold technique used? How did you determine that clear contamination had occurred? You assume the DRI CFA is the benchmark, and any large deviations that don't align with that record must be contamination? It currently reads as a bit subjective.

Thanks for this comment.

We now explain more in detail difficulties that occurred during the CEP measurements, and tried to clarify the reasons for and approach used in the CEP data corrections. The DRI dataset is assumed to be correct since working conditions were as they should be, i.e. the ice section had the regular size for routine CFA measurements at DRI. This was not the case at CEP since the available ice core section for CFA measurements there were at the smaller size limit (and some core sections were below the limit) to work with the CFA and to work without contamination from laboratory air.

Taking the advantage of having a second dataset measured under regular conditions (the DRI dataset) over a limited depth interval, we made a comparison of the two datasets in the depth interval covered by both of them, to get an idea whether the analyzed CEP data are reliable and useful for their application in this study. Note that further down in the CDM ice core where no DRI data were available no additional data were discarded from the CEP dataset, but we evaluated the reliability of NO3 and NH4 data measured in these core depths. We state this now clearly in the text.

The corresponding text in Section 2. (now Section 2.2, line 215) now reads as follows:

*… However, since the CDM ice core has only a 3-inch diameter, the ice available for the CFA analyses at CEP consisted only of a non-rectangular cross-section with maximum outer dimensions of 2.5 x 3.0 cm instead of the standard quadratic size of 3.5 x 3.5 cm for which the standard melt head at CEP is designed. Although a special, smaller melt head was constructed for the CDM analyses, it was not always possible to assure that the CFA melt water only came from an inner section of the ice material with no contact to the outer surfaces. This may have led to a higher risk of contamination of the inner sample melt water stream and with the smaller melt water flow available implied also a reduced analyte spectrum. Despite the undersized core section available for the CFA analyses at CEP, 86% of $NO_3^-$ and/or $NH_4^+$ raw data could be evaluated. To test the reliability of the CEP dataset, the nitrate profiles obtained at DRI and CEP (covering 97% in this depth range) were compared over the depth interval 45 to 86 m. Both datasets are in very good overall agreement, except for individual outliers in the CEP data.*

*After having additionally discarded very high peaks (concentrations above 700 ppb) in $NO_3^-$ values (1.5% of CEP data in the depth interval from 45 to 86 m), which were not present in the DRI dataset and could be attributed easily to contamination, mean $NO_3^-$ values over this depth interval were 263±281 ppb (CEP) and 255±231 ppb (DRI) (Fig. 4). The agreement is somewhat weaker for $NH_4^+$ likely because for this species only 80% of this depth range is covered by CEP measurements. After discarding additionally 8 % of the CEP $NH_4^+$ data between 45 and 86 m consisting of high $NH_4^+$ peaks (concentrations exceeding 190 ppb), which were not present in the DRI dataset, the mean $NH_4^+$ values of 101±110 ppb (CEP) and 95±99 ppb (DRI) were in good agreement. As a consequence of the better reliability, we base our discussion mainly on the $NO_3^-$ data. Below 86 m no additional data were discarded from the CEP $NO_3^-$ and $NH_4^+$ datasets. However, because no further single $NO_3^-$ peak values above 700 ppb were found below 86 m, we are confident in $NO_3^-$ data below this depth. In*

*the case of NH$_4$$^+$ we cannot exclude that a few peaks in the record below 86 m with a concentration higher than 200 ppb might be influenced by contamination.*

- Was the winter to summer layer thickness ratio obtained just from ammonium? Were other glaciochemistry time series used as well?

  Thanks for this comment, we should have written this more correctly in the text. The winter to summer layer thickness ratio is based as the annual layer dating, not just on NH4 but on different aerosol tracers which all show a seasonal variation. This is now reported in more detail in the text.

  The text in Section 3.1 (line 246) was corrected accordingly to:
  *… and the winter to summer layer thickness ratio (which was calculated using the seasonal information embedded in the various aerosol tracers measured in the core, Preunkert et al., 2000), decreases from 1 at the surface to ~0.5 at 100 m depth.*

  *Based on the well-marked seasonality in the chemical stratigraphy for all cores, annual layer counting was used as the main dating tool over the time period of interest of this study, i.e., back to the 1950s. NH$_4$$^+$ has a very strong seasonal variation (factor of ~14 higher in summer than in winter) caused by the parallel seasonal changes in source strengths and vertical transport of NH$_4$$^+$ (Preunkert et al., 2000). However also other ions (such as nitrate and sulfate) show clear seasonal variations (factor of ~4 higher in summer than in winter). The annual layer counting, which was based mainly on ammonia, was reinforced by absolute time markers such as Saharan dust events (for example the prominent event in 1977) (Preunkert et al., 2000 for C10; Legrand et al., 2013 for CDK, Legrand et al., 2018 and this study for CDM) and radiometric analyses aimed at detecting fallout from atmospheric thermonuclear bomb testing via $^3$H (Legrand et al., 2013 for CDK and this study for CDM) and $^{137}$Cs (Vincent et al., 1997) for C10. ..*

- "The dating of the C10 ice core back to 1925 obtained from annual layer counting of the ammonium record was initially established by Preunkert et al. (2000). More recently, the availability of additional measurements such as lead, cadmium and thallium allowed the dating to be extended back to 1890 without changing the original dating back to 1935 (Legrand et al., 2018)." This implies (with no mention of it) that you changed the original dating between 1925 and 1935, yes? Maybe say a bit more about this (how did you identify a dating error? etc).

  Thanks for this comment.
  In fact, when having measured lead (stable isotopes) and cadmium in the C10 ice core in 2016 (and two years later also in the lower layers of CDK), we recognized that the beginning of the industrial use of these metals in ~ 1890 was visible in the C10 (and CDK) core in form of a significant increase of these metals. With that we had an additional absolute time marker in 1890 and mainly the depth age relation between the last existing absolute time marker (the fluride increase in the beginning of the 1930s and this new time marker in 1890). The updated data

series were then used in publications made in the following, as stated in the text of the present manuscript.

Also, as also stated in the manuscript, that dating update changed only the very lower parts of the depth age relations, which were not in the focus of the present study. Therefore, we kept explanations about that topic rather short. On the other hand, for data consistency between the older publications and more recent ones (including the present one), we added a more precise explanation in the manuscript in Section 3.1.1 and also Section 3.1.2.

In addition, since Fig. 4 (now Fig. 5) shows data back to 1925 we updated it also (without changing its scientific meaning) to contain the actual data as they are archived in the database and not the originally published dataset versions of Legrand et al., 2013 and Preunkert et al., 2003.

The text in Section 3.1.1. (line 263) reads now:

*… More recently, new measurements of toxic metals such as lead, cadmium and thallium underpinned identification of an additional absolute time marker, visible as a marked concentration increase in all three metals at the beginning of the industrial period, what allowed to extend the C10 chronology back to 1890 (Legrand et al., 2018). This additional information did not significantly change the original dating back to 1935 (i.e., only by one year back to depth 106.5 m and 5 years at a depth of 112 m), and these changes are within the estimated dating uncertainty of 5 to 10 years (Preunkert et al., 2000).*

The text in Section 3.1.2. (line 298) now reads:

*… Analogous to C10 (see Sect. 3.1.1), the CDK dating was updated in the lower part on the basis of additional measurements of trace metals such as lead and cadmium without changing significantly the original dating of Legrand et al. (2013) back to 1935 (Preunkert et al., 2019a).*

[Figure]

**Figure 5: (a) Annual layer thickness of C10 (adapted from Preunkert et al. 2000), CDK (adapted from Legrand et al., 2013) and CDM. For CDM, the annual layer thickness is estimated via the ammonium stratigraphy back to 1980 and via the nitrate (and ammonium) stratigraphy further back in time (Sect. 3.1.3). (b) comparison of nitrate summer half-year means of C10 (adapted from Preunkert et al., 2003), CDK (adapted from Legrand et al., 2013) and CDM. The thick solid lines for C10 and CDK refer to the smoothed profiles (single spectrum analysis, see Legrand et al., 2013). CDM depth intervals for which the dating is uncertain (Sect. 3.1.3), are marked with dashed lines.**

- The bomb test horizon insets shown in Fig.4 are really lacking. I certainly understand only looking in certain sections of the core for bomb horizons (where they're expected), but why are there so few measurements? There are so few that it isn't actually possible to confidently pick 1963 (or <1954) at CDM or CDK. Were only wings (bag averages) available?

  Thanks for this comment, I think you meant Fig. 4 (now Fig. 5)?
  Commonly, 3H was measured on integrated samples (bag averages, as supposed by the reviewer), on 0.6 to 1 m length to identify the location of the 1963 peak and the start of the bomb tests. Such a sample resolution (covering around 1 to max 2 years) is "normally" enough to identify the 1963 peak and to localize the start of the bomb tests to ± 1 - 2 years. The expected depth of the horizons was determined via annual layer counting, so that we could avoid an overcharging of the capacity of the 3H lab. Note that the maximum of 1963 itself has "normally" a width of 2 years in rain (Global Network of Isotopes in Precipitation. *The GNIP Database* Accessible at: http://isohis.iaea.org) and snow (Schotterer et al., 1998) deposition records. Therefore, we feel, that if the peak in CDK would have been

included in the record we should have detected it. This was the only aim why the analyses were made.

**Reviewer 4**
This paper discusses dating anomalies in 3 cores drilled at different dates at Col du Dome. An earlier version of this paper has been commented on by two other reviewers. They made important points about the need to clarify the proposed mechanism. This is my first review of the paper, so I can see that significant improvements have been made, but I retain the concern of the previous reviewers. It is still written quite confusingly in places, and the proposed mechanism is still not really explained in a way that makes sense to me. Having said that, I do appreciate this focus on the difficulties of dating Alpine cores, and I also acknowledge the strength of using the evidence of missing layers along with the evidence of enhanced 210Pb to try and find an explanation. I think the paper can be published but would benefit from one further round of changes. It will still be a bit unconvincing and speculative but if well-explained, this paper can form the basis for a better appreciation of what is needed to use and date alpine cores well.
Specific comments (both detailed and general)

Line 40 "entirely cold". What does this mean. Do you mean it's polar and not temperate, ie has no meltwater? If so please spell this out.

Thanks for this comment. Yes, you are right cold meant here that there is no melting influence from the upper into deeper ice layers.

We updated the text as following (line 39):
*... Although it has experienced significant warming in response to climate change since the 1980s (Vincent et al., 2007; Gilbert and Vincent, 2013; Vincent et al., 2020), the glacier has been shown to be entirely cold (i.e. the ice temperature is below freezing point at all depths), with the exception of sporadic surface melting and refreezing in the uppermost centimeters during summer......*

Line 50 and onwards. The entire paper rests on us believing that it is possible to get a really good age scale from annual layer counting if the stratigraphy is continuous and that this is achieved for C10. This is asserted with reference particularly to Preunkert et al (2000). However, the casual reader is going to see the nitrate profiles shown in Figure 3 (C10 panel) and really wonder if this is possible. I appreciate that dating was mainly done with ammonium (which makes me wonder why this is not shown). Perhaps to reassure readers less familiar with the previous work the authors could include a plot similar to Fig 7 and 8 from Preunkert (2000) in the supplement, to really pin down the reliability of the C10 dating. This might be referred to at line 192 where the assertion hat C10 is well-dated is most clearly made.

Thanks for this comment.
The present study relyies on well-established work and scientific results obtained and published over the last 25 years. Therefore, we might have passed a bit too quickly over the dating of C10 for readers who are not familiar with the glaciochemistry of alpine ice cores, in writing that the "dating was achieved on the base of ammonia profile" what is a bit oversimplified since NH4 was not the only available tool to date.
On the other hand, we cannot discuss all the already established features in the C10 and CDK cores but need to reference existing publications for this. The dating on the base of ionic profiles, and hereby especially NH4 due to its stronger seasonal variations is

known in literature and is also used for ice cores from other high alpine sites such as Colle Gnifetti (Eichler et at. 2023), Fiescherhorn (Schwikowski,et al., 1999; Jenk et al., 2014), Grenzgletscher (Eichler et al., 2000).

The publication in which the dating of C10 is detailed (Preunkert et al., 2000) is on open access, and the interested reader can quickly discover the glaciochemical setting of the drill site, the detailed (at the top Figure 3 and 5 and further down Figure 7) and mean (Figure 4) seasonality of ions and deuterium as well as the depth age relationship (Figure 8) established in this epoch. However, to convince the reader unfamiliar with these older publications of the age scale we added another figure in this round of revisions. As update of this original depth age relation achieved in 2000, in view of additional absolute time markers which confirmed the dating, and to document the recent update made on the depth-age relation over the time period before 1935 (even if that period is not in the focus of the present study), we added a figure in the introduction of the manuscript reporting the actual depth-age relationship of C10. Along with the annual resolution, absolute time markers identified in C10 are assigned including 6 horizons situated between 1986 and 1930 with maximal distances of 15 years between each other.

[Figure]

**Figure 1: Depth-age relation established for C10 between 2000 and 2016. Data are from Preunkert et al., 2000, Vincent et al. 1997, Preunkert et al., 2001a, Preunkert et al., 2001b, Legrand et al., 2002, Legrand et al., 2018 and Preunkert et al., 2019a and the age scale is based on annual layer counting and absolute age markers. The shaded area refers to the depth zone where enhanced [210]Pb values (see Sect. 3.2 and Fig. 6) were observed.**

In addition, we added in Section 3.1 (line 251) more details about the seasonal variations of NH4, NO3 and SO4 to emphasize that there are, despite their different amplitudes, coherent in timing:

*Based on the well-marked seasonality in the chemical stratigraphy for all cores, annual layer counting was used as the main dating tool over the time period of interest of this study, i.e., back to the 1950s. $NH_4^+$ has a very strong seasonal variation (factor of ~14 higher in summer than in winter) caused by the parallel seasonal changes in source strengths and vertical transport of $NH_4^+$ (Preunkert et al., 2000). However also other ions (such as nitrate and sulfate) show clear seasonal variations (factor of ~4 higher in summer than in winter). The annual layer counting, which was based mainly on ammonia, was reinforced by absolute time markers such as Saharan dust events (for example the prominent event in 1977) (Preunkert et al., 2000 for C10; Legrand et al., 2013 for CDK, Legrand et al., 2018 and this study for CDM) and radiometric analyses aimed at detecting fallout from atmospheric thermonuclear bomb testing via $^3H$ (Legrand et al., 2013 for CDK and this study for CDM) and $^{137}Cs$ (Vincent et al., 1997) for C10.*

Sections 2 and 3. I found the structure, where some of the methods are folded into Section 2, very offputting. The authors are discussing the site characteristics and presenting a first summary of what they think might be happening (as requested by the editor), and then in a sudden gear change at line 112 they start discussing the analytical methods. I would strongly recommend splitting Section 2 into 2 main Sections: "Site characteristics", and "Ice core analysis". The analysis discussion could also be a bit more structured, perhaps even with a table, as I found the mass of different instruments and labs to be overwhelming as I read it. Section 3 is then simply two parts, "Dating" and "210Pb", not "Dating and Methods".

Thanks for this comment. We agree totally with you and split Section 2 in two parts. 2.1 Site characteristics and 2.2 Ice core analysis. Concerning the ice core analysis, we arranged the analysis now by parameters and not by lab as before. That should make this Section more readable. In addition, we added a table (Table 2) which reports analytical methods applied for samples used within the present study. Following your recommendations, the header of Section 3 was changed to "Dating and $^{210}Pb$ data".

Table 2 reads:

**Table 2: Analytical methods of ice core analysis used for the present study**

| Core name | C10 | CDK | CDM |
|---|---|---|---|
| $^{210}Pb$ | gamma-spectrometry (IGE) (two samples, this study; all others Vincent et al., 1997) | alpha-spectrometry (IUP, Legrand et al., 2013) | alpha-spectrometry (IUP, this study) |
| $^{137}Cs$ | gamma-spectrometry (IGE, Vincent et al., 1997) | - | - |
| $^3H$ | - | gas counting (IUP, Legrand et al., 2013)) | gas counting (IUP, this study) / liquid scintillation (CEP, this study) |
| $NO_3^-$ and $NH_4^+$ | ion chromatography (Preunkert et al., 2003; Fagerli et al., 2007)- | ion chromatography (Legrand et al., 2013)- | ion chromatography (IGE, Eichler et al., 2023) / continuous flow analyses (DRI, Legrand et al., 2018; CEP, this study) |

Table 1 is useful. The bracketed value for CDM 3H maximum is not explained – I assume it is the two alternatives.

Thanks for this comment. Yes, you are right, the value in parenthesis assigned the depth of the 3H peak which is considered as not corresponding to year 1963. We updated Table 1 with a footnote that reads:

*a) Depth of shallower $^3$H maximum detected in CDM, considered as not corresponding to the year 1963 (see Section 3.1.3)*

Line 222. This analytical detail should be with the earlier discussion of methods.

Thanks. The respective sentence was moved to Section 2.2.

Line 263 – this was very confusing. You refer to the "second peak (89.5 to 96 m depth)" but the two peaks are at 88 and 93 m. The values you discuss here of 10-40 TU are also not the peak values. Please reword this to explain exactly what you mean, clarifying what you mean by the "second peak", and which time period and depth you are comparing the 10-40 TU values to.

Similarly line 279 "If the first 3H peak" - which is that – the one encountered at shallower depth or the one that was deposited earlier (ie deeper)? Please review all text from lines 263 to 283 to ensure you are clear about what you are comparing. I am sure this can be expressed more clearly so the reader can follow the different options.

Thanks for this comment. Indeed, the wording was not very clear.

In line 263 of the previous version of the manuscript we wrote "around the second peak (89.5 to 96 m depth)" and meant the ice layers above and below the deeper 3H peak. We clarified this now in the text (this line (line 355) reads now:

*This is consistent with the observed value in CDM ice above and below the deeper $^3$H peak (which is around 93.5 m depth).*

and also changed the wording from "first" and "second" to "shallower" and "deeper", respectively in the complete paragraph.

Line 418 to 423. This is confusing. Surely the (partial) opening of the crevasse would most simply lead to a reduced buildup of 222Rn concentrations diffusing into the firn, and could therefore give any reduction you want (right down to background levels if completely open). I don't see why you need (even if you have) disturbed layers to get this reduction.

Thanks for your comment. You are right our conclusion was not entirely conclusive. We cancelled this argument and changed this part in Section 4 as following (line 581):

*... A partial opening of the crevasse to the atmosphere would allow the bedrock-derived $^{222}$Rn in the crevasse to mix with the much lower atmospheric $^{222}$Rn concentrations (Pourchet et al., 2000). This would have led to a strong reduction of additional $^{222}$Rn accumulation and $^{210}$Pb production in the crevasse and in the snow and firn around the crevasse, starting from the moment of the opening to the atmosphere. In addition, disturbed isochrones also could lead potentially to decreased $^{210}$Pb inventories since ice layers from the upstream side of the*

*crevasse with enriched $^{210}$Pb activities could be missing, as suggested for CDK but not CDM.*

Overall explanation. I think the authors have the elements of a solution but somehow the way they explain things doesn't quite work for me. I think this may be partly because neither I or they seem to be thinking in 3 dimensions. The problem for me is related to what they see as the character of the crevasse, because the question is how does ice get through it at all. If the crevasse only reaches partway through the depth of the ice sheet then presumably it flows with the glacier and would have reached the drill site by now. So the way the authors have drawn it with it open to the bed seems like the right way, but in that case no ice flows past it (assuming it is permanently open). So for any ice to reach our site it must be flowing round the crevasse and coming in at an angle to the flow line, and surely this is where the stratigraphy can get disturbed by taking a much longer flow path at some times than others. (I am not an ice dynamicist so perhaps this is wrong, but I would find it more convincing than what is written in lines 405-412 which seems to require ice to cross the crevasse).

Thanks for this remark. Our view of the functioning of the crevasse is now explained in detail in Section 2 (now Section 2.1, line 91):
*In fact, field observations and photographic evidence shows the existence of a large crevasse (clearly visible by a depression at the surface although the crevasse is not necessarily open to the atmosphere) east of the CDD dome which, dependent on its north-south extension, could also intersect the upstream flow line from the drill site. Unfortunately, we do not have direct measurements of the depth and lateral extent of the crevasse. The crevasse appears approximately at an oversteepening of the bedrock topography (Fig. 2c) along the flow line, suggesting that it is extensive stress at the bottom that leads to crack formation at a specific point and allows for opening of a deep crevasse down to bedrock (in line with $^{210}$Pb evidence as outlined below). We stress that the crevasse is not necessarily open to the surface (see Figure 2 a and b), but that collapse of the snow bridge at the top in the past cannot be ruled out. We also note that this crevasse is not moving downhill with the surface velocity of several meters per year in the observations but is found approximately at the same location of the glacier surface every year. Despite this stationarity of the crevasse, the surface velocity field is not disturbed (Gilbert et al. 2014) implying that the ice flow is not totally interrupted across the crevasse. Together with the stationarity of the crevasse, this suggests that the subsurface void created by the crevasse is filled again by glacier flow after its opening (as also suggested by significant glacier thickness reductions of a few meters from 1993 to 2017 (Vincent et al. 2020) in the vicinity of the crevasse). Accordingly, we interpret the glaciological evidence as (recurrent) opening but also potential re-closure of the crevasse (Colgan et al., 2016) below the surface at the same bedrock topography-induced position. Comparing photos taken in 2012 (Fig. 2a) and in 1999 (Fig.2b), shows widening and northward extension of the crevasse from 1999 to 2012. Whereas in 2012 the crevasse is clearly visible as a snow-covered depression on the surface slope, the crevasse appeared to be limited to the southwestern flank of the drill site catchment area in 1999. Following Fig. 3, the crevasse is situated more than 100 m upstream of the drill site of C10, CDK, and CDM. Figure 3a shows the CDD glacier thickness changes between 1993 and 2017 overlayed with the modelled flow line indicating the calculated arrival depths at the drill site of C10, CDK, and CDM (Gilbert et al., 2014). Figure 3b and c represent vertical cross sections along the modelled flow line in Fig. 3a overlayed by simplified sketches of the upstream crevasse visible in Fig. 2. We sketch the crevasse in two hypothesized temporal states, as concluded in Sect.*

*4 on the basis of C10, CDK and CDM ice core data presented in Sect. 3. Table 1 summarizes the main characteristics of the three ice cores and basic findings related to radiometric analyses.*

I do also get the argument about the 222Rn penetrating the firn, but some discussion of distance scales would be helpful here. 222Rn has a halflife of 3.8 days. I agree this means it can build up under a snow bridge, but it also surely implies that it can only penetrate horizontally into the firn by a few metres. This is then challenging for C10 where the authors require that the ice has not seen the crevasse. I would rather think that it must have seen the crevasse. I am then not sure how to explain the lack of flow disturbance. I don't have a solution but I think this should be discussed.

Thanks for your comment and thoughts. The fact that the crevasse has crossed the flowline already of C10 is evoked in the manuscript. in Section 4 (line 564):
*Although speculative, we assume that the upstream crevasse of Fig. 2 and 3 already existed earlier in the 1970s and was capped at the top by the snow bridge. If the crevasse (1) did not intersect the upstream flow line at the time the $^{210}$Pb anomaly was imprinted in the firn reaching C10 from the crevasse in 1994 (implying that the flow line was close enough to the crevasse for $^{222}$Rn to diffuse to the flow line) or (2) did intersect the catchment area of the C10 drill site at that time but was so narrow that the chronology of the C10 ice core was not significantly disturbed before closing again, ...*

Just before this we added a new paragraph in which a rough estimate is made concerning the distance over which the 222Rn charged air of the crevasse would influence the firn and the order of contact time needed to produce the $^{210}$Pb levels found at the drillsite.

It reads as follows in Section 4 (line 533):
*Pourchet et al. (2000) observed mean $^{222}$Rn activities of ~10,000 up to nearly 150,000 Bq m$^{-3}$ in the firn of Mont Blanc (2 km from Dome de Gouter with the same rock mineralogy) and 0.7 Bq kg$^{-1}$ of $^{210}$Pb on average in a depth of 0.5 m in the annual snow/firn layer of the measurement year, both of which are orders of magnitude higher than typical background values in air or snow. They attributed these high levels to Rn emanation from a nearby crevasse. Vincent reported as much as 28 Bq kg$^{-1}$ in a firn/ice core at the summit of Dome de Gouter which had contact with a subsurface crevasse. Since snow accumulation and ice flow velocities at the Dome de Gouter are lower than at the Mont Blanc summit, we assume that the firn air of the core drilled at Dome de Gouter was in much longer contact with the crevasse than the surface snow layer at Mont Blanc summit.*

*In the following we make a rough estimate whether such $^{222}$Rn activities are sufficient to explain the $^{210}$Pb anomaly in our cores. To keep the estimation simple, we assume a temporal constant $^{222}$Rn activity in the crevasse of 50,000 Bq m$^{-3}$, which lies in the typical range observed by Pourchet et al 2000 and is equivalent to a $^{222}$Rn number concentration of 2.4 $10^{10}$ m$^{-3}$ in the crevasse. As the half life of $^{210}$Pb is much longer than that of $^{222}$Rn we can assume that the amount of $^{210}$Pb loaded into the firn is controlled primarily by the total $^{222}$Rn entering the firn. We assume that $^{222}$Rn loads the adjacent firn by diffusion and that after ten $^{222}$Rn half lives the radiogenic $^{222}$Rn entering the firn has essentially decayed to zero limiting its entrainment length to a few meters (see diffusion length discussion above). As a*

*first order estimate we assume a linear concentration gradient between the $^{222}$Rn concentration in the crevasse and zero radiogenic $^{222}$Rn in a distance of a diffusion length after ten $^{222}$Rn half lives. We acknowledge that the true concentration gradient is not linear and that some $^{222}$Rn atoms will enter deeper into the firn than this diffusion length, but to obtain the order of magnitude loading of the firn with $^{210}$Pb this back-of-the-envelope calculation seems justified. Using the $^{222}$Rn concentration in the crevasse as a measure of the concentration gradient driving the diffusive flux, this leads to a diffusive $^{222}$Rn atom flux into the firn of 12,000 and 40,000 m$^{-2}$ s$^{-1}$ for a firn diffusivity of 0.1 10$^{-5}$ m$^{-2}$ s$^{-1}$ and 1 10$^{-5}$ m$^{-2}$ s$^{-1}$, respectively. At an exposure length of the bedrock of one year and a firn density of 500 kg m$^{-3}$, this would result in a loading of the firn by approximately 800 to 2,500 10$^{6}$ $^{210}$Pb atoms kg$^{-1}$, equivalent to an initial $^{210}$Pb activity on the order of 800 to 2500 mBq kg$^{-1}$. Given that the ice flows within 1 to 2 $^{210}$Pb half lives from the crevasse to the ice core drill site, these numbers are of the same order of magnitude as the activities measured in the firn core. We note that an exposure time of the crevasse of one year may be at the upper limit of what is possible (given the stationary position of the crevasse which requires healing of the crevasse before ice flow has moved its position significantly) and an exposure time of only one month may not be sufficient to explain the measured activities. On the other hand, we only based our estimate on diffusive entrainment of $^{222}$Rn into the firn. If there also are pressure differences between the crevasse air and the firn air (for example induced by synoptic pressure variations at the surface) this would also lead to an advective flux into the firn which may increase the initial $^{210}$Pb activity after $^{222}$Rn loading of the firn. In summary, while a more precise estimate would require stringent firn transport modeling in and around the crevasse which is beyond the scope of this paper, the overall order of magnitude of the $^{210}$Pb anomaly can be explained by our simple estimate.*

Additional literature not referenced in the manuscript:
Schwikowski, Margit & Brütsch, S. & Gaeggeler, Heinz & Schotterer, Ulrich. (1999). A high-resolution air chemistry record from an Alpine ice core: Fiescherhorn glacier, Swiss Alps. Journal of Geophysical Research. 1041. 13709-13720. 10.1029/1998JD100112.

Jenk, Theo & Szidat, S. & Schwikowski, Margit & Gaeggeler, Heinz & Brütsch, S. & Wacker, L. & Synal, H.-A & Saurer, Matthias. (2006). Radiocarbon analysis in an Alpine ice core: Record of anthropogenic and biogenic contributions to carbonaceous aerosols in the past (1650-1940). Atmospheric Chemistry and Physics. 6. 10.5194/acpd-6-5905-2006.

SchottererU.,StichlerW.,GrafW.BürkiH-U.,Gourcy L., Ginot P. and Huber T. (1998), Stable isotopes in Alpine ice cores: Do they record climate variability, Techniques in the Study of Environmental Change, IAEA.

IAEA/WMO (2006). Global Network of Isotopes in Precipita- tion. *The GNIP Database* Accessible at: http://isohis.iaea.org, retrieved 11/2008